# HYDRA: Model Factorization Framework for Black-Box LLM Personalization

**Yuchen Zhuang[1], Haotian Sun[1], Yue Yu[1], Rushi Qiang[1], Qifan Wang[2], Chao Zhang[1], Bo Dai[1]**

[1] Georgia Institute of Technology, [2] Meta AI

{yczhuang, haotian.sun, yueyu, rqiang6, chaozhang}@gatech.edu,
wqfcr@fb.com, bodai@cc.gatech.edu

## Abstract

Personalization has emerged as a critical research area in modern intelligent systems, focusing on mining users' behavioral history and adapting to their preferences for delivering tailored experiences. Despite the remarkable few-shot capabilities exhibited by black-box large language models (LLMs), the inherent opacity of their model parameters presents significant challenges in aligning the generated output with individual expectations. Existing solutions have primarily focused on prompt design to incorporate user-specific profiles and behaviors; however, such approaches often struggle to generalize effectively due to their inability to capture shared knowledge among all users. To address these challenges, we propose HYDRA, a model factorization framework that captures both user-specific behavior patterns from historical data and shared general knowledge among all users to deliver personalized generation. In order to capture user-specific behavior patterns, we first train a reranker to prioritize the most useful information from top-retrieved relevant historical records. By combining the prioritized history with the corresponding query, we train an adapter to align the output with individual user-specific preferences, eliminating the reliance on access to inherent model parameters of black-box LLMs. Both the reranker and the adapter can be decomposed into a base model with multiple user-specific heads, resembling a hydra. The base model maintains *shared* knowledge across users, while the multiple personal heads capture *user-specific* preferences. Experimental results demonstrate that HYDRA outperforms existing state-of-the-art prompt-based methods by an average relative improvement of 9.01% across five diverse personalization tasks in the LaMP benchmark. Our implementation is available at https://github.com/night-chen/HYDRA.

## 1 Introduction

Pre-trained large language models (LLMs) [35, 36, 2, 31, 4, 30] have revolutionized various natural language processing (NLP) tasks, ranging from traditional recommendation systems [62, 12, 8, 6] to modern virtual assistants [9, 23, 63, 66, 44, 65, 47]. Despite their strong capabilities, LLMs require further customization to consistently demonstrate desirable behaviors to each user and achieve optimal performance in specific use cases [48, 43, 61, 10, 13, 14, 24, 57, 58, 45, 59]. As a result, LLM personalization has emerged as a rapidly evolving area of research [20], with the goal of tailoring the emergent abilities of LLMs to meet the unique needs of individual users.

Several existing studies have shown effectiveness in personalizing LLMs, including (1) fine-tuning personalized LLMs for each user [50, 54] and (2) aligning LLMs to personalized preferences through Reinforcement Learning from Human Feedback (RLHF) [22, 55, 17]. However, both fine-tuning and RLHF-based methods require access to model parameters, restricting their use to white-box LLMs only (e.g., LLaMA-2 [51]). These models tend to be less capable than black-box LLMs (e.g., GPT-3.5

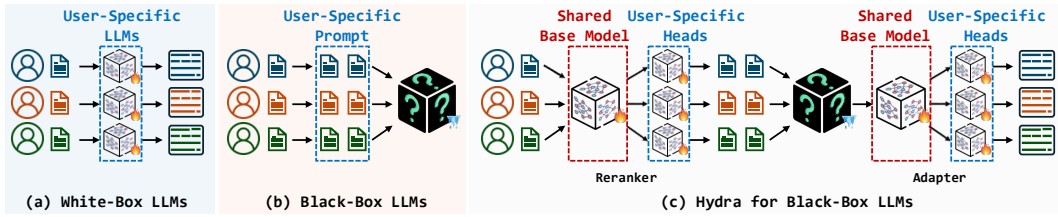

Figure 1: Personalization in white-box and black-box LLMs. Existing methods prioritize (a) fine-tuning user-specific models in white-box LLM personalization, while (b) designing user-specific prompts for black-box LLM personalization. In HYDRA, we present (c) a learning-based model factorization solution to enhance the effectiveness of personalization in black-box LLMs. 🔥 indicates the trainable parameters, whereas ❄️ indicates the inaccessible fixed parameters.

[30]) because they have access to less training data and smaller model scales. Moreover, RLHF-based methods require more explicitly attributed characteristics (*e.g.*, style) [22, 17] from implicit user behavior history and necessitate excessive annotation efforts for capturing human preferences.

Without access to modify the model parameters for black-box LLM personalization, an alternative solution is to augment user-specific content and/or context into the prompt template. One straightforward approach is to incorporate the user's complete profile or entire historical behavior into the prompt design [6, 18, 25, 52, 64, 5, 64]. However, integrating the entire profile may exceed the length limitations of LLMs and lead to substantial costs, while randomly selected records cannot effectively capture representative patterns. To address this dilemma, retrieval-augmented generation (RAG) approaches [41, 40, 29, 21] have been explored by extracting the most relevant information from the user's historical data to facilitate personalized generation. One limitation is that a retrieval-augmented framework encodes different users independently in a personalized prompt, making it challenging to capture the shared (global) patterns of the entire user group [50]. Moreover, in comparison to fine-tuning the entire or partial model parameters to create personalized language models for individual users [50, 22, 17], simply augmenting the input prompt through a centralized LLM without updating model parameters may diminish the effectiveness of personalization (Figure 1).

To address these challenges, we propose HYDRA, a learning-based model factorization framework that captures both user-specific and shared behavior patterns to enable effective personalization within black-box LLMs. HYDRA leverages a retrieval-augmented workflow, where a retriever initially extracts relevant user behaviors from historical data for effective user-specific preference identification. To achieve personalized generation, we focus on the training process of two fundamental components: (1) a personalized reranker to prioritize useful user information from the retrieved records, and (2) a personalized adapter to align black-box LLM outputs with user-specific preferences, without requiring access to internal model parameters. Both the reranker and the adapter can be decomposed into a base model with multiple personalized heads, similar to a Hydra. By employing model factorization, we effectively integrate shared (global) knowledge, captured by the centralized base model, with user-specific preferences, harnessed through multiple user-specific heads, to enhance generalization across the entire user group.

We conduct extensive experiments on LaMP [41], a comprehensive language model personalization benchmark, to evaluate the personalization capabilities of HYDRA across multiple dimensions, including three text classification tasks and two text generation tasks. Notably, HYDRA achieves an average improvement of 4.8% over the best-performing baselines across all five diverse tasks. Further in-depth studies reveal the robust capability of HYDRA in scaling up to accommodate larger user groups and extensive behavior history, as well as adapting to user behavior shifts. We also demonstrate the effectiveness of shared knowledge in enhancing user experience through both quantitative and qualitative analyses. To facilitate future research in black-box LLM personalization, we will release the code repository and model checkpoints for transparency and reproducibility.

Our main contributions are as follows: (1) We propose HYDRA, a black-box LLM personalization framework that effectively mines user behavior history and adapts to user preferences for enhanced user experience; (2) HYDRA integrates shared (global) knowledge from the base model and individual (local) preference from multiple user-specific heads through model factorization to deliver generalizable personalization; and (3) HYDRA significantly outperforms existing prompt-based methods across five diverse tasks in the LaMP benchmark [41], introducing a novel learning-based solution that achieves more effective adaptation to individual users in black-box LLMs.

## 2 Related Works

**In-Context Learning.** Vanilla personalized prompting approaches leverage the powerful in-context learning capability of LLMs by using the user's randomly sampled behavior history as contextual samples. Existing studies [6, 18, 25, 52, 64] have utilized encoding user histories-whether personal ratings, interaction histories, or exemplary reviews-as few-shot examples to facilitate LLMs in generating personalized content in various down-stream applications. Additionally, research has shown that utilizing a longer user history can potentially lead to better performance [5, 64].

**Profile-Augmented Prompting.** Improving upon the random sample strategy and leveraging the insights from the enhanced performance with more historical information, profile-augmented generation (PAG) summarizes user preferences and behavior patterns into natural language profiles for query augmentation. For instance, Richardson et al. [38] employ instruction-tuned LLMs to generate abstract summaries of user history data, integrating summarization for enhanced personalization. Similarly, ONCE [26] creates user profiles by summarizing topics and regions of interest from their browsing history, thereby assisting LLMs in capturing user preferences for downstream tasks.

**Retrieval-Augmented Prompting.** Compared to random sampling in in-context learning and the use of entire histories in PAG, retrieval-augmented prompting excels at extracting the most relevant records from user behavior history to enhance LLM personalization, thereby efficiently managing the growing user behavior data within LLMs' limited context length and supporting personalized generation with more relevant evidence. For instance, LaMP [41, 40] introduces a retrieval-augmented method to obtain the most relevant content from the user's behavioral history and incorporate it into the prompt design. Similarly, AuthorPred [21] retrieves relevant past user-written documents for personalized text generation. Pearl [29] proposes a generation-calibrated retriever to select from historic user-authored documents, enhancing personalization with relevant prompt augmentation.

**Limitations.** Similar to in-context learning, personalization based on PAG is prone to be easily distracted by irrelevant information retrieved, especially when there is a shift in user behavior between the current query and the user's historical records. Despite the improvement upon PAG by retrieving relevant information, RAG-based methods may still suffer from the quality of retrieved information, where the "most relevant" information may not necessarily be the "most useful" information to answer a new query. Additionally, prompting-based methods not only lack deeper personalization analysis due to their reliance on a single centralized model but also lack access to global knowledge due to the user-specific prompt design.

## 3 HYDRA: Model Factorization for Black-Box LLM Personalization

### 3.1 Problem Formulation: Black-Box LLM Personalization

Black-box LLM personalization refers to tailoring model generations to align with user preferences according to their history [20, 50, 17], without having access to model parameters. Specifically, given a black-box LLM $G$ and a training dataset $\mathcal{D}_{\text{train}} = \{(q_u, r_u, \mathcal{H}_u)\}$, where for each user $u$, $q_u$ indicates the input sequence, $r_u$ refers to the target output, and $\mathcal{H}_u$ represents the user's historical behavior containing preference information. The user history data $\mathcal{H}_u = \{h_u^i\}$ includes all user behaviors $h_u^i$, consisting of $(q_u^i, r_u^i)$ pairs, mirroring the task-specific query-answer format $(q_u, r_u)$. The goal of personalization is to adapt the LLM's generation $\hat{r}$ to the target output $r$, conditioned on both the input and the user history.

Traditional fine-tuning methods train universal models for all users, whereas personalized tuning aims to develop a unique model for each user $u$, denoted as $\theta^{(u)}$, which captures the unique characteristics of its user-specific data distribution $\mathcal{D}^{(u)}$. Ideally, each user model should possess both *shared* (general) and *user-specific* knowledge. Thus, the problem can be formulated as a decomposition of $\theta^{(u)}$ into a shared part $\sigma$ and a personalized part $\tau^{(u)}$ to learn general and user-specific knowledge, respectively. The training object of LLM personalization can be formulated as:

$$\min_{\sigma, \tau^{(1)}, \tau^{(2)}, \cdots, \tau^{(u)}} \sum_{u=1}^{U} \mathcal{L}_u(\sigma; \tau^{(u)}; \mathcal{D}^{(u)}), \tag{1}$$

where $\mathcal{L}_u(\sigma; \tau^{(u)}; \mathcal{D}^{(u)})$ denotes the loss function of user $u$, and $U$ is the total number of users. Specifically, in the black-box LLM personalization, the model parameters in $G$ are not accessible,

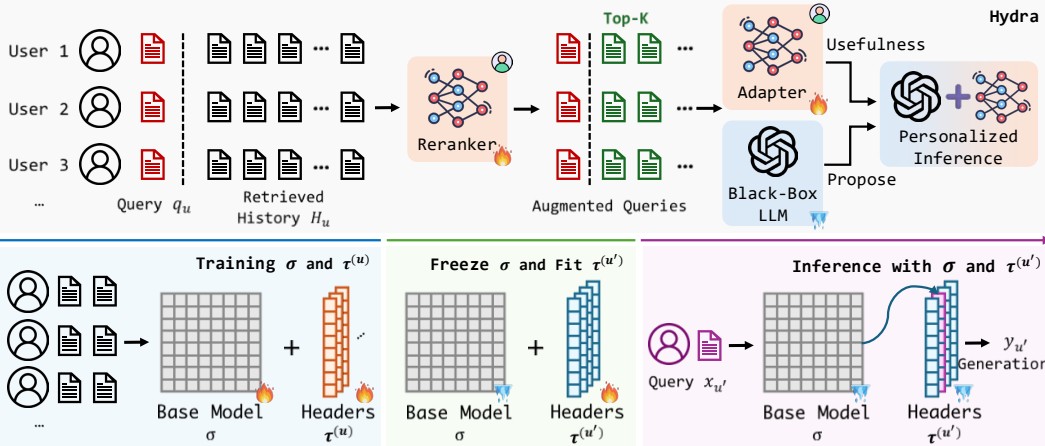

Figure 2: Overview of HYDRA. HYDRA follows a retrieval-augmented framework: (1) Firstly, we extend an original RAG to a two-stage *retrieve-then-rerank* workflow, where we rerank the most useful information from relevant user behavior records to capture user-specific preference (Section 3.2); (2) Secondly, augmented by the selected historical data, we train an adapter to align the output of black-box LLMs with personalized human preference (Section 3.3). Both the reranker and the adapter can be decomposed into a base model with multiple user-specific heads, resembling a hydra-like structure (Section 3.4). The base model maintains *shared* knowledge across users, while multiple personal heads capture *user-specific* preferences. ⊗ represents the model decomposition for personalization.

making it infeasible to directly fine-tune the black-box LLM. Thus, we present HYDRA, a model factorization framework for black-box LLM personalization, as shown in Figure 2.

## 3.2 Retrieve-then-Rerank

To capture user-specific preference, we follow a *retrieve-then-rerank* workflow to (1) retrieve the relevant user behavior records and (2) rerank them based on usefulness. Specifically, given an input query $x$, we employ a retriever to retrieve top-$N$ user history behaviors that have the highest relevance scores when compared with the input $x$. The objective of the reranker is to identify user historical records that can serve as useful user preference information for answering the user's incoming queries.

**Training Data Generation.** Within the training data, each user only has a single query, which is insufficient for capturing the relationships between the behaviors of the same user. Scoring all the history pairs of users is quadratic in $\sum_{u=1}^{|\mathcal{D}|} |\mathcal{H}_u| \times |\mathcal{H}_u|$, which becomes prohibitively time-consuming. In order to obtain a high-quality candidate set of training samples for each user query $q_u$, we retrieve the top-$M$ relevant history records, denoted as $\mathcal{R}(q_u, \mathcal{H}_u)$. Additionally, to gain a better understanding of the user's history, we randomly sample another $M$ historical records that can be considered as the user's previous queries. For each of these previous queries, we also retrieve the top-$M$ relevant historical records. The training candidates can be represented as:

$$\mathcal{E}_{\text{reranker}} = \{(q_i, r_i, h_i)\}_{i=1}^{|\mathcal{E}_{\text{reranker}}|} = \{\{(q_u, r_u, \mathcal{R}(q_u, \mathcal{H}_u))\} \cup \{(h_u^i, \mathcal{R}(h_u^i, \mathcal{H}_u/h_u^i))\}_{i=1}^M\}_{h_u^i \in \mathcal{H}_u u \in \mathcal{D}},$$

(2)

where $q_i$ indicates the query, $r_i$ indicates the ground-truth answer to the query, and $h_i$ indicates the candidate history. We then utilize an LLM as a labeling function to measure the potential improvement (*i.e.*, usefulness) of each history upon the LLM personalized generation. Specifically, for each candidate $(q_i, r_i, h_i)$, we first sample the generation of LLM $\hat{r}_i$ by using the input context $q_i$ and the candidate history $h_i$ from $p_{\text{LLM}}(\hat{r}_i | q_i, h_i)$. Next, we compare the generation $\hat{r}_i$ with the ground-truth answer $r_i$, and create a binary label $y_i = \mathbb{1}(\hat{r}_i = r_i)$. For generation task, the condition is soften to $y_i = \mathbb{1}(\hat{r}_i \approx r_i)$, where the Rouge metrics between $r_i$ and $\hat{r}_i$ is above a pre-defined threshold. To optimize the reranker, we combine the user query with the candidate history in an entailment learning style, resulting in the training inputs $x_i = \{[\text{CLS}] \, q_i \, [\text{SEP}] \, h_i \, [\text{SEP}]\}$. Therefore, we can curate a training set for the reranker $\mathcal{D}_{\text{reranker}} = \{(x_i, y_i)\}_{i=1}^{|\mathcal{D}_{\text{reranker}}|}$.

HYDRA**-Reranker Training.** Training the reranker for each user allows for a personalized selection of relevant historical information based on the user's query. The training objective involves using the cross-entropy loss between the predictions made by HYDRA-Reranker heads and the ground truth. For a specific user $u$ and the corresponding training sample $(x_i^u, y_i^u) \in \mathcal{D}_{\text{reranker}}$, we can calculate the cross-entropy loss function for HYDRA-Reranker as follows:

$$\mathcal{L}_{\text{HYDRA}}^{\text{reranker}} = -y_i^u \log p_i^u - (1 - y_i^u) \log(1 - p_i^u), \tag{3}$$

where $p_i^u$ indicates the model prediction made by HYDRA-Reranker.

HYDRA**-Reranker Inference.** With the trained reranker, our objective is to select the top-$k$ most relevant candidates from the retrieved user history for a personalized generation. For each query $q_i$ from user $u$ in the test data, we retrieve the most relevant history $\overline{\mathcal{H}}_i^u = \{h_{i,1}^u, \cdots, h_{i,N}^u\}$ and concatenate them to form the test inputs $\{x_{i,1}^u, \cdots, x_{i,N}^u\}$. By feeding these into HYDRA-Reranker, we obtain the corresponding predictions $\{p_{i,1}^u, \cdots p_{i,N}^u\}$. From the test inputs, we select the top-$k$ history candidates $\mathcal{C}_i$ with the highest level of usefulness:

$$\mathcal{C}_i = \arg \text{top-}k_{h_{i,j} \in \overline{\mathcal{H}}_i}(p_{i,j}^u). \tag{4}$$

### 3.3 Black-box LLM Adaptation

Complementary to personalizing the few-shot demonstrations for LLMs in Section 3.2, we further align the black-box LLM generation to user-specific preference through the training of a personalized adapter, eliminating the need for directly accessing model parameters.

**Training Data Generation.** To take full advantage of the preference information hidden in the user query and history, we generate the candidates for the adapter training:

$$\mathcal{E}_{\text{adapter}} = \{(q_i, r_i)\}_{i=1}^{|\mathcal{E}_{\text{adapter}}|} = \{q_u, r_u\}_{u \in \mathcal{D}} \cup \{\{(q_u^i, r_u^i)\}_{i=1}^{|\mathcal{H}_u|}\}_{u \in \mathcal{D}}. \tag{5}$$

For each input query $q_i$ from the training dataset, we augment the reranked history candidates $\mathcal{C}_i$ with the query and sample $b$ candidate responses from the black-box LLM:

$$\{\hat{r}_{i,j}\}_{j=1}^b \sim p_{\text{LLM}}(\hat{r}_i | \mathcal{C}_i, q_i). \tag{6}$$

In comparison with the ground-truth personalized generation $r_i$, we can evaluate each generated solution $\hat{r}_{i,j}$ and assign a corresponding binary label $y_i$ as $y_i = \mathbb{1}(\hat{r}_{i,j} = r_i)$. Using the model generations, we establish a new dataset for the adapter training, denoted as $\mathcal{D}_{\text{adapter}} = \{x_{i,j}, y_i\}_{i=1}^{|\mathcal{D}_{\text{adapter}}|}$, where $x_{i,j} = \{[\text{CLS}] \ q_i \ [\text{SEP}] \ \hat{r}_{i,j} \ [\text{SEP}]\}$ represents the concatenation of the user query with the entire candidate generation.

HYDRA**-Adapter Training.** We utilize the model decomposition training strategy (detailed in Section 3.4) to train personalized adapters for each user, which allows us to tailor the selection of candidate generation that best aligns with the user's preference. To calculate the task-specific loss function, we follow Eq. (3) and employ the same cross-entropy loss between the predictions of user-specific heads and the ground truth to optimize HYDRA-Adapter.

HYDRA**-Adapter Inference.** During the process of model inference, we conceptualize the black box LLM as a proposal generator, while the adapter functions as an evaluator. Specifically, for each test query $x_i$ from user $u$, we adopt the best-of-$b$ inference, also known as rejection sampling. Initially, we sample $b$ candidate generations $\{\hat{r}_{i,j}\}_{j=1}^b$ from the LLM. By passing them through the HYDRA-Adapter, we obtain the corresponding score for each candidate $\{p_{i,1}^u, \cdots, p_{i,b}^u\}$. The solution with the highest score is then chosen as the final answer:

$$\hat{r}_i^u = \arg \max_{j=1,\cdots,b} p_{i,j}^u. \tag{7}$$

### 3.4 Model Factorization for Personalization

To further effectively capture the shared knowledge across all users as well as the individual user-specific behavior patterns, a direct solution is to develop a parameter decomposition method, where the shared parameters store the general knowledge, benefiting from a high model capacity, while the remaining parameters are designed to learn user-specific preferences that complement the overall

understanding (typically on a smaller scale). Hence, it is adequate to employ a smaller-sized model (adapter) to represent these personalized parameters, instead of fine-tuning the entire LLM.

**Model Factorization.** Formally, in HYDRA, we assume that each personalized language model is associated with a set of weights $\Theta = \{\theta^{(u)}\}_{u \in \mathcal{D}}$, where $\theta^{(u)}$ represents the weights of the personalized model for user $u$. Each $\theta^{(u)}$ is decomposed as:

$$\theta^{(u)} = \sigma \otimes \tau^{(u)}, \tag{8}$$

where $\sigma$ represents the base model parameter matrix that is shared among all users, $\tau^{(u)}$ represents a personalized head model, and $\otimes$ indicates the operation of appending the user-specific head $\tau^{(u)}$ on top of the base model $\sigma$. Specifically, we add $|\mathcal{D}|$ heads to the final hidden states $s$ generated by the original model. The $u$-th head measures the usefulness between query and histories, or the level of preference for the generation from the perspective of the $u$-th user. The prediction of the $u$-th head is denoted as $p^{(u)}$. For each head, we employ a single layer of feed-forward network. We define $u$-th head as $\tau^{(u)} = [\mathbf{W}_1^{(u)}, \mathbf{W}_2^{(u)}, \mathbf{b}_1^{(u)}, \mathbf{b}_2^{(u)}]$ and the prediction can be outlined as:

$$p^u = \text{softmax}(\mathbf{W}_2^{(u)} \cdot \text{Tanh}(\mathbf{W}_1^{(u)} \cdot s + \mathbf{b}_1^{(u)}) + \mathbf{b}_2^{(u)}), \text{ where } \mathbf{W}_2^{(u)} \in \mathbb{R}^{d \times o}, \mathbf{W}_1^{(u)} \in \mathbb{R}^{d \times d}, \tag{9}$$

where $d$ indicates the dimension of hidden states and $o$ indicates the dimension of outputs.

**Training Strategy.** We then train the base model in conjunction with multiple user-specific heads. For each sample $(x_i^u, y_i^u)$ of user $u$ from the training data and the model prediction $p_i^u$, we update the base model and the $u$-th head accordingly:

$$\sigma \leftarrow \sigma - \alpha \nabla \mathcal{L}(y_i^u, p_i^u), \ \tau^{(u)} \leftarrow \tau^{(u)} - \alpha \nabla \mathcal{L}(y_i^u, p_i^u), \tag{10}$$

where $\alpha$ indicates the learning rate and $\mathcal{L}$ represents the task-specific loss function.

**Fit Test User History.** In order to accommodate the newly incoming users in the test data, we cannot reuse the personalized heads from the training data. Therefore, we need to create and initialize new heads. To fit the test users' history, we freeze the base model and solely focus on the heads. This fitting process is simple and requires minimal computational resources. Similar to the update of $\tau^{(u)}$ in Eq. (10), when given a new user $u'$ for testing, we also leverage the task-specific loss function $\mathcal{L}(\cdot)$ to update the head $\tau^{(u')}$:

$$\tau^{(u')} \leftarrow \tau^{(u')} - \alpha \nabla \mathcal{L}(y_i^{u'}, p_i^{u'}), \tag{11}$$

where $(x_i^{u'}, y_i^{u'})$ are obtained from test user history.

**Personalized Inference for Test Users.** Upon fitting the test user history, we obtain the base model $\sigma$, which contains general knowledge across users, and the user-specific head $\tau^{(u')}$, which captures the user-specific knowledge in the test user history. Therefore, we can apply personalized inference directly to the test user $u'$, given its query $x_{u'}$:

$$p_i^{u'} = f_{\sigma \otimes \tau^{(u')}}(x_i^{u'}), \tag{12}$$

where $f_{\sigma \otimes \tau^{(u')}}(\cdot)$ indicates the inference of the model $\sigma \otimes \tau^{(u')}$.

# 4 Experiments

## 4.1 Experimental Setup

**Datasets and Tasks.** We adopt a widely used language model personalization benchmark, LaMP [41], focusing on a diverse set of personalized text classification and generation tasks, including (1) Personalized News Categorization (**LaMP-2N**), (2) Personalized Movie Tagging (**LaMP-2M**), (3) Personalized Product Rating (**LaMP-3**), (4) Personalized News Headline Generation (**LaMP-4**), and (5) Personalized Scholarly Title Generation (**LaMP-5**). For data splitting, we follow the user-based separation setting provided by the LaMP benchmark, with 100 randomly selected users for training and an additional 50 randomly selected users for testing. No shared users are displayed across splits for specific measurements in personalization for new users. Additional details of the LaMP benchmark are available in Appendix D.

Table 1: Main experiment results on the LaMP benchmark. R-1 and R-L represent ROUGE-1 and ROUGE-L, respectively. $k$ indicates the number of retrieved items. ↑ denotes that higher values are better, while ↓ implies that lower values are preferred. The best score and 2nd best score for each task are highlighted in **bold** and underlined, respectively. Notations are consistent across tables.

| Dataset (→) | LaMP-2N | | LaMP-2M | | LaMP-3 | | LaMP-4 | | | LaMP-5 | | |
|---|---|---|---|---|---|---|---|---|---|---|---|---|
| Method (↓) | Acc. ↑ | F-1 ↑ | Acc. ↑ | F-1 ↑ | MAE ↓ | RMSE ↓ | R-1 ↑ | R-L ↑ | BLEU ↑ | R-1 ↑ | R-L ↑ | BLEU ↑ |
| `gpt-3.5-turbo` [30] | 0.660 | 0.280 | 0.440 | 0.309 | 0.480 | 0.825 | 0.133 | 0.120 | 0.996 | 0.379 | 0.326 | 5.477 |
| ICL-Random (k=1) | 0.640 | 0.293 | 0.480 | 0.360 | 0.660 | 0.990 | 0.165 | 0.144 | 1.567 | 0.410 | 0.349 | 5.327 |
| ICL-Random (k=2) | 0.640 | 0.284 | 0.520 | 0.400 | 0.560 | 0.917 | 0.171 | 0.158 | 1.923 | 0.413 | 0.348 | 6.698 |
| ICL-Random (k=4) | 0.660 | 0.288 | 0.480 | 0.356 | 0.560 | 0.917 | 0.163 | 0.148 | 1.589 | 0.393 | 0.349 | 7.445 |
| RAG (k=1) [41] | 0.680 | 0.293 | 0.400 | 0.290 | 0.500 | 0.837 | 0.151 | 0.130 | 1.417 | 0.418 | 0.353 | 5.852 |
| RAG (k=2) [41] | 0.600 | 0.281 | 0.460 | 0.343 | 0.580 | 0.906 | 0.173 | 0.156 | 1.778 | 0.419 | 0.367 | 6.898 |
| RAG (k=4) [41] | 0.640 | 0.284 | 0.460 | 0.340 | 0.580 | 0.970 | 0.172 | 0.156 | 1.812 | 0.415 | 0.362 | 7.382 |
| PAG (k=0) [38] | 0.640 | 0.293 | 0.500 | 0.356 | 0.520 | 0.894 | 0.163 | 0.150 | 1.724 | 0.410 | 0.359 | 6.124 |
| PAG (k=1) [38] | 0.680 | 0.308 | 0.520 | 0.362 | 0.560 | 0.913 | 0.170 | 0.156 | 1.674 | 0.397 | 0.327 | 6.481 |
| HYDRA-Reranker (k=4) | 0.760 | 0.375 | 0.520 | 0.393 | 0.480 | 0.775 | 0.177 | 0.166 | 1.951 | 0.423 | 0.368 | 6.864 |
| HYDRA-Adapter | 0.680 | 0.277 | 0.480 | 0.385 | 0.420 | 0.762 | 0.145 | 0.118 | 1.137 | 0.409 | 0.355 | 5.816 |
| HYDRA | **0.780** | **0.401** | **0.540** | **0.458** | **0.400** | **0.747** | **0.178** | **0.169** | **2.396** | **0.434** | **0.372** | **7.531** |

Table 2: Ablation studies of HYDRA. -P. represents removing the HYDRA personalized training strategy.

| Dataset (→) | LaMP-2N | | LaMP-2M | | LaMP-3 | | LaMP-4 | | | LaMP-5 | | |
|---|---|---|---|---|---|---|---|---|---|---|---|---|
| Method (↓) | Acc. ↑ | F-1 ↑ | Acc. ↑ | F-1 ↑ | MAE ↓ | RMSE ↓ | R-1 ↑ | R-L ↑ | BLEU ↑ | R-1 ↑ | R-L ↑ | BLEU ↑ |
| HYDRA | **0.780** | **0.401** | **0.540** | **0.458** | **0.400** | **0.747** | **0.178** | **0.169** | **2.396** | **0.434** | **0.372** | **7.531** |
| -P.-Adapter | 0.740 | 0.300 | 0.500 | 0.384 | 0.460 | 0.831 | 0.169 | 0.157 | 1.604 | 0.398 | 0.336 | 5.766 |
| -P.-Reranker | 0.700 | 0.298 | 0.500 | 0.379 | 0.480 | 0.825 | 0.162 | 0.154 | 2.063 | 0.399 | 0.347 | 5.364 |
| -P.-Adapter & Reranker | 0.780 | 0.312 | 0.420 | 0.339 | 0.520 | 0.894 | 0.172 | 0.163 | 2.001 | 0.385 | 0.332 | 5.848 |
| HYDRA-Reranker (k=4) | 0.760 | 0.375 | 0.520 | 0.393 | 0.480 | 0.775 | 0.177 | 0.166 | 1.951 | 0.423 | 0.368 | 6.864 |
| -P.-Reranker | 0.700 | 0.358 | 0.480 | 0.372 | 0.540 | 0.864 | 0.174 | 0.161 | 1.772 | 0.411 | 0.363 | 6.192 |
| HYDRA-Adapter | 0.680 | 0.277 | 0.480 | 0.385 | 0.420 | 0.762 | 0.145 | 0.118 | 1.137 | 0.409 | 0.355 | 5.816 |
| -P.-Adapter | 0.673 | 0.276 | 0.460 | 0.341 | 0.480 | 0.835 | 0.141 | 0.119 | 1.006 | 0.374 | 0.317 | 4.754 |

**Baselines.** We compare our proposed HYDRA with existing state-of-the-art black-box LLM personalization methods, including in-context learning (**ICL-Random**, with random selected $k$-item from user behavior history), retrieval-augmented prompting (**RAG**) [41], and profile-augmented prompting (**PAG**) [38]. We also present the experimental results of `gpt-3.5-turbo` [30] (zero-shot) without profile/history augmentation in order to showcase the baseline performance of the backbone language model. Learning-based personalization, such as fine-tuning and RLHF-based methods, cannot be applied to black-box LLMs (*e.g.*, `gpt-3.5-turbo`) due to the unavailability of model parameters. Details of baseline implementations are available in Appendix E.

**Evaluation Metrics.** Following the evaluation metrics specified in LaMP [41], we utilize accuracy (Acc.) and F-1 score (F-1) for the test classification tasks in LaMP-2N and LaMP-2M. Additionally, we employ mean absolute error (MAE) and root mean squared error (RMSE) for the ordinal multi-class classification task in LaMP-3. To comprehensively evaluate the personalized text generation tasks in LaMP-4 and LaMP-5, we report the ROUGE-1 (R-1), ROUGE-L (R-L), and BLEU metrics.

**Implementation Details.** For both baselines and our proposed HYDRA, we follow the same prompt template in the LaMP benchmark [41] and employ `gpt-3.5-turbo(1106)` and BM25 [39] as the backbone black-box LLM and default retriever, respectively. Additionally, both HYDRA-Reranker and HYDRA-Adapter leverage the lightweight `LongFormer-Base` (110M) [1] as the backend language models. We include more implementation details in Appendix F.

## 4.2 Main Results

Table 1 presents the main experimental results of five varied personalization tasks in the LaMP benchmark. Compared to the zero-shot setting in `gpt-3.5-turbo` [30], even a random selection of historical records from user behavior or profiles enhances the model performance in most tasks, suggesting that personalization contributes to improved performance with black-box LLMs. `HYDRA` exhibits substantial relative improvements across all five tasks, with an average of 9.01% over the best-performing baselines. Specifically, `HYDRA` outperforms the best-performing alternative by 14.71% in accuracy for the Personalized News Categorization (LaMP-2N) task, 3.85% in accuracy for the Personalized Movie Tagging (LaMP-2M) task, 20.00% in MAE (where lower values indicate better performance) for the Personalized Product Rating (LaMP-3) task, 2.89% in R-1 for the Personalized News Headline Generation (LaMP-4) task, and 3.58% in R-1 for the Personalized Scholarly Title Generation (LaMP-5) task. This further improvement can be attributed to the improved quality (usefulness) of retrieved records in better representing user-specific preferences and the effective integration of global knowledge sourced from the entire user group.

## 4.3 Ablation Studies

From Table 1, both `HYDRA`-Reranker and `HYDRA`-Adapter demonstrate their effectiveness by significantly improving the RAG and PAG baselines separately. Furthermore, we observe that `HYDRA` outperforms each of them individually, showcasing the complementary role of both components. In Table 2, we further eliminate the personalized training strategy of `HYDRA`, deteriorating to training a singular model across the entire user group, which solely incorporates global knowledge and lacks individualized customization for each user. The decline in model performance across all tasks highlights the significance of personalization through user-specific heads. In addition, we conduct additional experiments to comprehensively evaluate different components of `HYDRA` in black-box LLM personalization, including the effect of retrievers (Appendix G.4), the effect of adapters (Appendix G.5), and the effect of reranker with case studies and error analysis (Appendix H).

## 4.4 Scale-up Analysis

**Number of Users in Training.** We study the impact of users in the training data, as depicted in Figure 3 (a)-(c). As the number of users increases from 20 to 100, we observe that `HYDRA` reaches over 90%

**Number of History per User.** We examine the impact of the number of historical records per user in Figure 3 (d)-(f). We randomly chose 50 users from each range of number of history per user. As the number of historical records increases, we consistently observe improved performance in both classification and generation tasks, as indicated by all metrics. This demonstrates the robustness of `HYDRA` in effectively capturing user-specific preferences, especially with a larger amount of user historical data.

**Number of Selected History ($k$) per User.** In Figure 3 (g)-(i), we analyze the impact of the selected items $k$ per user. It can be observed that `HYDRA`

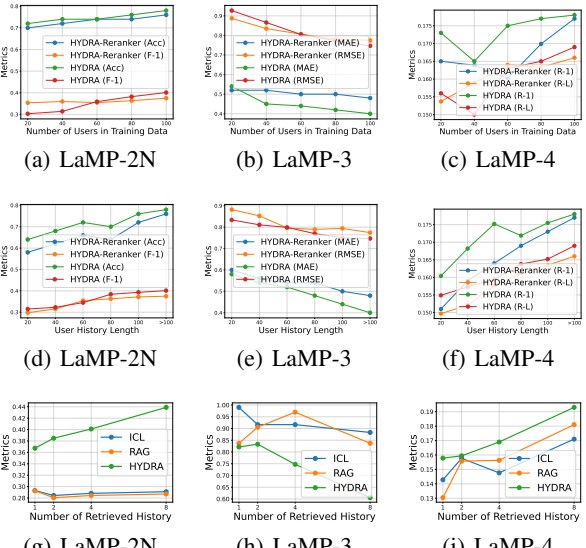

Figure 3: Sacle-up analysis for (a)-(c) Effect of users in training data, (d)-(f) Effect of historical records per user, (g)-(i) Effect of selected historical records ($k$) per user.

consistently outperforms other baselines across all values of $k$, highlighting the robustness of `HYDRA`. Additionally, it is evident that model performance correlates with the number of retrieved historical records in most cases, providing further evidence of the effectiveness of the retrieval-augmented

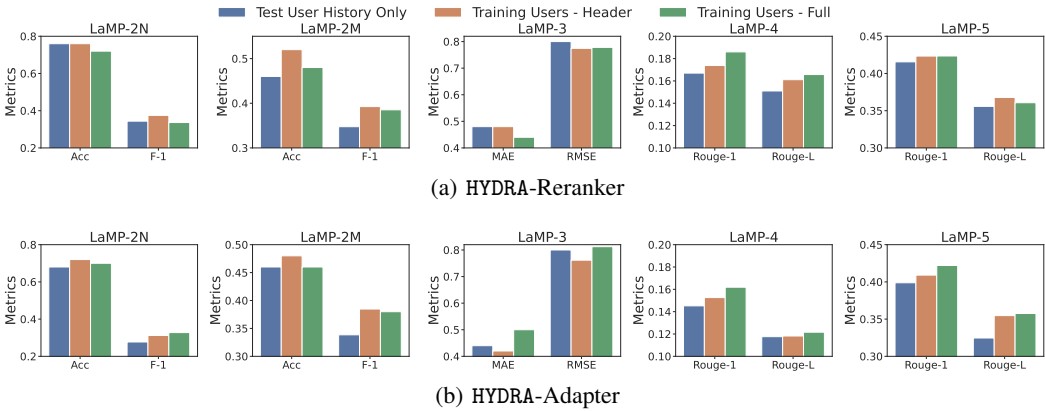

(a) HYDRA-Reranker

(b) HYDRA-Adapter

Figure 4: Performance under different levels of shared knowledge.

| Dataset ($\rightarrow$) | LaMP-2N | | LaMP-2M | | LaMP-3 | | LaMP-4 | | | LaMP-5 | | |
|---|---|---|---|---|---|---|---|---|---|---|---|---|
| Method ($\downarrow$) | Acc. $\uparrow$ | F-1 $\uparrow$ | Acc. $\uparrow$ | F-1 $\uparrow$ | MAE $\downarrow$ | RMSE $\downarrow$ | R-1 $\uparrow$ | R-L $\uparrow$ | BLEU $\uparrow$ | R-1 $\uparrow$ | R-L $\uparrow$ | BLEU $\uparrow$ |
| gpt-3.5-turbo | 0.638 | 0.499 | 0.412 | 0.347 | 0.540 | 0.851 | 0.133 | 0.119 | 1.043 | 0.439 | 0.371 | 6.018 |
| ICL-Random (k=1) | 0.598 | 0.476 | 0.392 | 0.335 | 0.676 | 0.959 | 0.147 | 0.132 | 1.330 | 0.457 | 0.396 | 8.118 |
| ICL-Random (k=2) | 0.632 | 0.499 | 0.376 | 0.311 | 0.562 | 0.871 | 0.151 | 0.137 | 2.388 | 0.451 | 0.393 | 8.550 |
| ICL-Random (k=4) | 0.630 | 0.518 | 0.392 | 0.352 | 0.440 | 0.740 | 0.161 | 0.146 | 2.418 | 0.457 | 0.396 | 8.404 |
| RAG (k=1) | 0.610 | 0.486 | 0.408 | 0.345 | 0.602 | 0.871 | 0.154 | 0.138 | 1.649 | 0.468 | 0.405 | 7.820 |
| RAG (k=2) | 0.624 | 0.479 | 0.380 | 0.315 | 0.559 | 0.836 | 0.161 | 0.149 | 2.958 | 0.480 | 0.419 | 9.021 |
| RAG (k=4) | 0.656 | 0.524 | 0.392 | 0.339 | 0.391 | 0.716 | 0.167 | 0.155 | 3.615 | 0.479 | 0.418 | 9.108 |
| PAG (k=0) | 0.618 | 0.489 | 0.404 | 0.340 | 0.583 | 0.872 | 0.161 | 0.141 | 1.950 | 0.460 | 0.405 | 7.372 |
| PAG (k=1) | 0.630 | 0.500 | 0.418 | 0.357 | 0.414 | 0.787 | 0.163 | 0.153 | 2.934 | 0.474 | 0.414 | 8.372 |
| HYDRA | 0.748 | 0.551 | 0.446 | 0.373 | 0.328 | 0.656 | 0.175 | 0.167 | 4.772 | 0.508 | 0.442 | 9.519 |

Table 3: Scale-up experiment results on the LaMP benchmark, including 1000 and 500 users during training and testing, respectively.

framework in black-box LLM personalization. However, an inconsistency is noted with RAG in LaMP-3, which may be attributed to the presence of noisy or irrelevant retrieved history. This finding emphasizes the importance of retrieval quality and raises the significance of the HYDRA-Reranker in measuring the usefulness of retrieved items. Additional scale-up experimental results are available in Appendix G.2.

**Number of Users in Training Set.** We also conduct additional scale-up experiments (Table 3) to evaluate HYDRA with an increased number of users, increasing from 100 to 1000, across all five tasks. Our findings from the scale-up experiments show that HYDRA maintains its performance advantages over baselines as the number of users increases.

## 4.5 Empirical Personalization Analysis

**Performance w/o Shared Knowledge.** We conduct additional experimental analysis on both text classification and generation tasks for a comprehensive evaluation. In Figure 4, we study how the shared knowledge is leveraged in HYDRA. Specifically, we create three settings with different levels of leveraging shared knowledge: (1) using *only test user history* for training, (2) updating just the *head* parameters using training user data, and (3) updating the *full* model parameters using training user data. We observe that incorporating a larger volume of training user data yields superior results compared to solely relying on test user history. The inclusion of additional training user information aids in capturing global information effectively. In addition, updating the entire model shows inferior performance compared to updating only the heads. This discrepancy arises due to the increased number of parameters, which necessitates a larger amount of data to accurately capture each user's

information. Additionally, the transition from training user information to testing user information results in a domain shift of global knowledge.

**Performance under User Behavior Shift.** Following the setup in Tan et al. [50] (see Appendix G.3 for additional details), we evaluate HYDRA under user behavior shift when the query is not relevant to all historical records. As illustrated in Figure 5, our proposed HYDRA continues to outperform the current state-of-the-art baselines in the face of user behavior shift. This

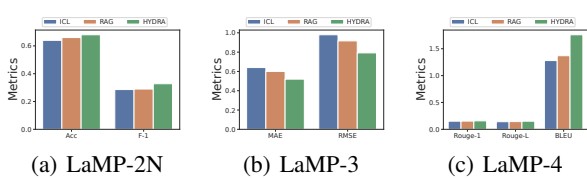

(a) LaMP-2N      (b) LaMP-3      (c) LaMP-4

Figure 5: Performance under user behavior shift.

serves as evidence of its robustness and generalizability in providing a widely applicable solution for black-box LLM personalization. This improvement can be attributed to the personalized reranker, which reevaluates the candidates based on their utility rather than solely on relevance, together with the personalized adapter further adjusts to align with user preferences.

**Effect of Significant Disparities in User Behavior.** To consider more extreme cases of disparities in user behavior, we retrain HYDRA on a mixture of 50% users with the fewest interactions and another 50% users with the most interactions (blue in Figure 6, compared to the random selection in orange). The black and gray dashed lines represent the best-performing baselines on the first and second metrics. The experimental results demonstrate that HYDRA consistently outperforms existing baselines, even under extreme cases. Compared to the previous random selection of training users, HYDRA achieves relatively lower performance due to the imbalance of training samples for dense users and sparse users. By leveraging the global information in shared parameters, knowledge can be effectively transferred from dense users to sparse users, thereby enabling further personalization through the utilization of sparse user-specific head models.

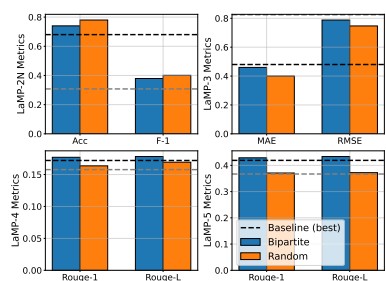

Figure 6: Experiments on LaMP benchmark of significant disparities in user behavior history.

## 5   Conclusions

In this paper, we proposed HYDRA, a model factorization framework for black-box LLM personalization that captures and leverages both user-specific and shared behavior patterns from historical data. Compared to prompt-based methods, HYDRA introduces a novel learning-based paradigm for black-box LLM personalization that not only remains highly effective in personalizing outputs by training a reranker with a black-box LLM adapter, but also eliminates the need for access to model parameters, presenting a promising alternative to existing mainstream techniques. Experimental results demonstrate that HYDRA outperforms state-of-the-art RAG and PAG baselines by an average relative improvement of 9.01% across five diverse personalization tasks. HYDRA establishes a foundation for learning-based black-box LLM personalization, facilitating targeted enhancements in user experience and ultimately contributing to the development of human-centric intelligent systems.

## Acknowledgments and Disclosure of Funding

We thank the anonymous reviewers and area chairs for their valuable feedback. This work was supported by NSF grants IIS-2008334, IIS-2106961, CAREER IIS-2144338, IIS-2403240, and Azure credits from the Microsoft Accelerating Foundation Models Research Program.

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

# A  Limitations and Broader Impacts

## A.1  Limitations

In this work, we propose a learning-based framework `HYDRA` for black-box LLM personalization. Despite its effectiveness, we have identified several potential limitations of `HYDRA`:

**Resource Limitations.** Due to limited access to the API services of black-box LLMs and budget constraints, our experiments with black-box fine-tuning are currently restricted to utilizing the Microsoft Azure OpenAI API service with the `gpt-3.5-turbo(1106)` model.

**Data Privacy.** Unlike the Microsoft Open AI fine-tuning service for black-box LLMs, which involves data exchange through API services, `HYDRA`-Adapter does not share any training data with third parties via APIs. It ensures the security of the training samples during black-box LLM adaption. However, it is important to note that potential private information may be contained in the retrieved historical records or the query itself, which could introduce a potential risk of private data leakage [60].

**On-Device Inference.** In real-world recommendation systems or virtual assistants, on-device inference with small-scale language models may be necessary [56]. While `HYDRA`-Adapter is a lightweight language model with 110M parameters and is efficient in fine-tuning during black-box LLM adaptation, the extensive model parameters of black-box LLMs themselves may eliminate the possibility of on-device inference.

## A.2  Broader Impacts

**Potential Positive Societal Impacts.** The proposed `HYDRA` addresses an important challenge posed by the inherently opaque nature of state-of-the-art LLMs, like GPTs [30], enabling learning-based personalization in black-box LLMs. Effective personalization of black-box LLMs can lead to significant improvements in user experience and engagement across a wide range of applications, from virtual assistants to recommendation systems. By tailoring the outputs of LLMs to the unique preferences and behavioral patterns of individual users, `HYDRA` has the potential to provide more useful, relevant, and satisfying interactions. This could enhance productivity, decision-making, and overall quality of life. Moreover, the ability to personalize language models without requiring access to their internal parameters makes this technology more accessible. This democratization of personalization capabilities can benefit a broader segment of the population, including individuals and organizations that may not have the resources or technical expertise to fine-tune or adapt large language models directly. Increased accessibility to personalized language models could lead to more inclusive and equitable technological solutions.

**Potential Negative Societal Impacts.** However, the personalization of language models also carries some potential risks and negative societal impacts, including concerns around user privacy and data security [20], as the personalization process requires collecting and storing sensitive user behavioral data. Improper handling or misuse of this data could lead to privacy breaches and unauthorized profiling of individuals. Robust data governance frameworks and stringent privacy safeguards will be crucial to mitigate these risks. Additionally, the personalization of language models could potentially exacerbate issues of algorithmic bias and filter bubbles. If the personalization process is not carefully designed to avoid reinforcing existing biases or limiting users' exposure to diverse perspectives, it could contribute to the creation of echo chambers and the perpetuation of biased decision-making.

**Future Works.** Our research highlights several meaningful avenues for future exploration. It is imperative to prioritize fairness, diversity, and inclusivity in the personalization process in order to address privacy-related concerns. Furthermore, one potential direction is to expand the scope of our applications beyond LLM personalization to encompass multi-modal vision language model personalization. This endeavor will not only enhance user-centric intelligent systems but also contribute to elevating the overall user experience.

Overall, the proposed `HYDRA` framework has the potential to deliver significant positive societal impacts through enhanced user experience and increased accessibility to personalized language models. However, careful consideration of privacy, data security, and algorithmic bias mitigation will be necessary to address the potential negative consequences and ensure the responsible development and deployment of this technology.

## A.3 Ethical Statements

We strictly follow data usage guidelines during the interaction with Microsoft Azure's Open AI API service. Although we utilize all publicly available datasets in this study, we have withdrawn from the human review process by completing and submitting the Azure OpenAI Additional Use Case Form[1] to prevent any potential information leaks.

# B  Additional Related Works

## B.1  LLM Personalization

We summarize existing LLM personalization studies in Table 4, including (1) prompting-based methods (available for both white- and black-box LLMs), (2) learning-based methods (only available for white-box LLMs), and (3) HYDRA, a learning-based black-box LLM personalization framework.

Table 4: Summary of LLM personalization baselines and HYDRA on the inclusion of different components. We present an overview of the existing LLM adaptation methods, focusing on six key aspects: (1) personalization for specific users, (2) global knowledge across different users, (3) retrieval from user history, (4) retrieval for relevance, (5) retrieval for usefulness, (6) learning-based method and (7) personalization of black-box LLMs.

| Methods | Personalization | Global Knowledge | Retrieval | Retrieval Relevance | Retrieval Usefulness | Learning | Black-Box LLM |
|---|---|---|---|---|---|---|---|
| *Prompting-based Methods* | | | | | | | |
| ICL-Random | ✗ | ✗ | ✗ | ✗ | ✗ | ✗ | ✓ |
| RAG [40] | ✓ | ✗ | ✓ | ✓ | ✗ | ✗ | ✓ |
| PAG [10] | ✓ | ✗ | ✓ | ✓ | ✗ | ✗ | ✓ |
| *Learning-based Methods* | | | | | | | |
| Fine-Tuning | ✗ | ✓ | ✗ | ✗ | ✗ | ✓ | ✗ |
| OPPU [50] | ✓ | ✗ | ✗ | ✗ | ✗ | ✓ | ✗ |
| RLPHF [17] | ✓ | ✗ | ✗ | ✗ | ✗ | ✓ | ✗ |
| HYDRA (**Ours**) | ✓ | ✓ | ✓ | ✓ | ✓ | ✓ | ✓ |

## B.2  Learning-based Personalization for White-Box LLMs

**Aligning Language Models to Personal Preferences.** While RLHF aligns LLMs with general human preferences [33, 67, 46, 37], it is inadequate for capturing diverse and individual perspectives, as it assumes a reward model based only on average annotator preferences and ignores variations in user-desired outputs even for the same prompt [3, 42]. The personalized-RLHF (P-RLHF) framework [22] addresses diverse user preferences encoded in human feedback by jointly training a user model to capture individual preferences and a reward model to generate personalized language. Similarly, fine-grained RLHF [55] enhances personalization by providing rewards after each generated segment and integrating multiple reward models for various feedback types, including information incorrectness, irrelevance, and incompleteness. Additionally, Reinforcement Learning from Personalized Human Feedback (RLPHF) [17] further considers scenarios of conflicting preferences and achieves personalized alignment by decomposing preferences into multiple dimensions. However, RLHF-based solutions often focus on preference alignment by explicitly decomposing preferences into multiple predefined dimensions specified by the user, which may not always be feasible in real-world scenarios.

**Personalized Parameter-Efficient Fine-Tuning (PEFT).** An alternative approach to personalization is to develop individual language models tailored to each user [50, 54]. One notable example of relevant work, although only available on white-box LLMs, is One PEFT Per User (OPPU) [50], which trains a PEFT model for each user using Low-Rank Adaptation (LoRA) [14]. OPPU effectively combines parametric user knowledge embedded in personalized PEFT parameters with

---

[1]https://aka.ms/oai/additionalusecase

non-parametric knowledge obtained through retrieval and user profiling, ensuring a more comprehensive and personalized user experience.

**Limitations.** Learning-based methods primarily focus on white-box LLM personalization, involving personalized alignment through parameter merging and personalized reward models, which is not applicable to black-box LLMs due to the opacity of their model parameters. Specifically, RLHF-based solutions require explicit personalization information (e.g., difficulty, style) as preference pairs in addition to behavior history, which is difficult to collect and necessitates additional human effort in annotations. Compared with existing personalization solutions that develop individual user models for user-specific alignment, HYDRA takes into consideration shared (global) knowledge for more effective personalization with potential generalization to the entire user group. In addition, personalization is a strategic priority for many corporate corporations, where most state-of-the-art solutions remain proprietary and are not available as open-source.

## B.3 Black-Box LLM Adaptation

Table 5: Summary of LLM adaptation baselines and HYDRA-Adapter on the inclusion of different components. We present an overview of the existing LLM adaptation methods, focusing on five key aspects: (1) accessibility of model parameters, (2) availability of high-dimensional representations for input sequences or output generations, (3) availability of token probabilities, (4) necessity of retrieval corpus, and (5) utilization of a smaller adapter model. Adapted from Table 1 in Sun et al. [48].

| Methods | w/o Model Parameters | w/o High-Dimensional Representation | w/o Token Probabilities | w/o Retrieval Corpus | w/ Smaller Adapter |
|---|---|---|---|---|---|
| *White-Box LLM Fine-Tuning* | | | | | |
| Fine-Tuning [7] | ✗ | ✗ | ✗ | ✓ | ✗ |
| Instruction-Tuning [53] | ✗ | ✗ | ✗ | ✓ | ✗ |
| Continual Pre-Training [10] | ✗ | ✗ | ✗ | ✓ | ✗ |
| Adapter [13] | ✗ | ✗ | ✗ | ✓ | ✓ |
| Prefix-Tuning [27] | ✗ | ✗ | ✗ | ✓ | ✓ |
| LoRA [14] | ✗ | ✗ | ✗ | ✓ | ✓ |
| *Grey-Box LLM Adaptation* | | | | | |
| LMaaS [49] | ✓ | ✗ | ✗ | ✓ | ✓ |
| kNN-Adapter [15] | ✓ | ✓ | ✗ | ✗ | ✓ |
| CombLM [32] | ✓ | ✓ | ✗ | ✓ | ✓ |
| Proxy-Tuning [24] | ✓ | ✓ | ✗ | ✓ | ✓ |
| *Black-Box LLM Adaptation* | | | | | |
| BBox-Adapter [48] | ✓ | ✓ | ✓ | ✓ | ✓ |
| **HYDRA-Adapter (Ours)** | ✓ | ✓ | ✓ | ✓ | ✓ |

Following Sun et al. [48], we leverage the level of access to LLM internal parameters to determine its classification as white-box, grey-box, or black-box LLMs (Table 5). White-box LLMs grant full access to both model parameters and output probabilities, whereas grey-box models provide access only to output probabilities. In contrast, black-box models restrict access to both parameters and output probabilities.

Fine-tuning APIs, such as the OpenAI GPT-3.5-turbo API [34], enable the effective adaptation of black-box LLMs by allowing users to upload training data and download fine-tuned outputs. However, black-box LLM adaptation through APIs presents several significant challenges [48]: (1) **Transparency**: The fine-tuning process is largely opaque, with limited visibility into key aspects such as the extent of trainable layers and specific model weights. This lack of transparency hinders optimal customization, as users are restricted to adjusting a narrow set of hyperparameters, such as the number of tuning epochs. (2) **Privacy**: Uploading training data via APIs raises concerns about potential privacy breaches, particularly in sensitive domains. (3) **Cost**: Fine-tuning APIs incur substantially higher costs compared to inference, making the adaptation process expensive. Moreover, the cost of fine-tuning escalates significantly when hyperparameter tuning is involved, further increasing the financial burden. As a result, we propose HYDRA-Adapter to eliminate the requirement for direct access to model parameters or API services in order to enable black-box LLM adaptation.

# C   Algorithm Details

---

**Algorithm 1: HYDRA.**

---

**Input:** $\mathcal{D}_{\text{train}} = \{(q_u, r_u, \mathcal{H}_u)\}$: training data; $u$: User; $q_u$: the user's query; $r_u$: the ground-truth answer to the user query $q_u$; $\mathcal{H}_u$: the user $u$'s history behaviors; $R(q_u, \mathcal{H}_u)$: the retriever to retrieve top-$M$ relevant history records; $E$: training epochs;

// HYDRA-*Reranker*

Construct the training, test history, and test query candidates $\mathcal{E}_{\text{reranker}}^{\text{train}}, \mathcal{E}_{\text{reranker}}^{\text{history}}, \mathcal{E}_{\text{reranker}}^{\text{query}}$ via Eq.(2)

Adopt LLM as labeling function to create and annotate the dataset $\mathcal{D}_{\text{reranker}}^{\text{train}}, \mathcal{D}_{\text{reranker}}^{\text{history}}, \mathcal{D}_{\text{reranker}}^{\text{query}}$

**for** $e = 1, \cdots, E$ **do**
  Compute the loss function via Eq.(3) over $\mathcal{D}_{\text{reranker}}^{\text{train}}$
  Update the shared parameters and user-specific heads via Eq.(10)

Freeze the shared parameters in the base model of HYDRA-reranker

Reinitialize the user-specific heads of HYDRA-reranker

**for** $e = 1, 2, \cdots, E$ **do**
  Compute the loss function via Eq.(3) over $\mathcal{D}_{\text{reranker}}^{\text{history}}$
  Update the user-specific heads via Eq.(11)

Obtain model inference $\mathcal{C}_i$ on $\mathcal{D}_{\text{reranker}}^{\text{query}}$ via Eq.(12).

// HYDRA-*Adapter*

Construct the training, test history, and test query candidates $\mathcal{E}_{\text{adapter}}^{\text{train}}, \mathcal{E}_{\text{adapter}}^{\text{history}}, \mathcal{E}_{\text{adapter}}^{\text{query}}$ via Eq.(5)

Augment the user query $q_i$ with reranked history candidates $\mathcal{C}_i$ and sample $b$ candidate responses from the black-box LLM via Eq.(6)

**for** $e = 1, \cdots, E$ **do**
  Compute the loss function via Eq.(3) over $\mathcal{D}_{\text{adapter}}^{\text{train}}$
  Update the shared parameters and user-specific heads via Eq.(10)

Freeze the shared parameters in the base model of HYDRA-adapter

Reinitialize the user-specific heads of HYDRA-adapter

**for** $e = 1, 2, \cdots, E$ **do**
  Compute the loss function via Eq.(3) over $\mathcal{D}_{\text{adapter}}^{\text{history}}$
  Update the user-specific heads via Eq.(11)

Obtain model inference $\hat{r}_i$ on $\mathcal{D}_{\text{adapter}}^{\text{query}}$ via Eq.(12).

**Output:** The personalized generation $\hat{r}_i^u$ for the given query $q_i^u$ from user $u$.

---

# D   Dataset and Task Details

We employ the Language Model Personalization (LaMP) benchmark [41], an open-source benchmark specifically designed to train and evaluate the capability of language models in generating personalized content. LaMP encompasses a diverse set of tasks (with LaMP-2 comprising two tasks, LaMP-2N and LaMP-2M), covering both personalized text classification and generation tasks. We present the statistics of the datasets in Table 6 to provide a clear depiction of the dataset structure. We further offer detailed descriptions of each task as follows.

- **Task 1: Personalized News Categorization (LaMP-2N)** is a categorical text classification task that involves classifying news articles into one of 15 categories based on a journalist. Given an article written by a user, the model predicts its category using the user's history of articles and their corresponding categories.

- **Task 2: Personalized Movie Tagging (LaMP-2M)** is an ordinal text classification task that involves predicting one of 15 tags for a movie based on a user's tagging history. LaMP-2M evaluates the capability of a language model to assign tags to a movie description using the user's historical movie-tag pairs.

- **Task 3: Personalized Product Rating (LaMP-3)** is a text classification task that involves predicting product ratings as a five-class text classification problem. The objective is to predict an integer rating from one to five for a given review based on users' historical review and rating pairs. This task assesses the capability of language models to capture user-specific rating preferences.

- **Task 4: Personalized News Headline Generation (LaMP-4)** is a text generation task that involves generating personalized news headlines for given articles based on the authors'

historical article-title pairs. This task evaluates the language model's ability to replicate the authors' stylistic nuances in headline generation.

- **Task 5: Personalized Scholarly Title Generation (LaMP-5)** is a text generation task that involves generating titles for input articles based on the author's historical article and title pairs. This task extends the exploration of personalized text generation into scholarly domains, explicitly focusing on generating titles for research articles.

LaMP-6 has been excluded since the dataset is not publicly available. Furthermore, two additional tasks (LaMP-1 and LaMP-7) have been excluded from our empirical evaluation due to the inconsistent formats between user history and query.

- **Personalized Citation Identification (LaMP-1)** is a binary text classification task that involves predicting citation choices. It requires determining which of two candidate papers will be cited in a new paper based on a user's historical publication data. This process utilizes user profiles that include the titles and abstracts of their previous works. This task aims to evaluate a language model's capability to identify the citation preferences specific to each user.

- **Personalized Tweet Paraphrasing (LaMP-7)** is a text generation task that involves paraphrasing an input tweet into a personalized one, using the user's historical tweet data for stylistic guidance. This task evaluates a model's proficiency in reproducing the unique stylistic patterns in authors' social media posts.

Table 6: Dataset statistics of five different personalization tasks in the LaMP benchmark [41].

| Task | Type | # Train | # Validation | # Test | Input Length | Output Length | # Profiles | # Classes |
|---|---|---|---|---|---|---|---|---|
| LaMP-2N | Classification | 5914 | 1052 | 1274 | $65.40 \pm 12.29$ | - | $306.42 \pm 286.65$ | 15 |
| LaMP-2M | Classification | 5073 | 1410 | 1557 | $92.39 \pm 21.95$ | - | $86.76 \pm 189.52$ | 15 |
| LaMP-3 | Classification | 20000 | 2500 | 2500 | $145.14 \pm 157.96$ | - | $188.10 \pm 129.42$ | 5 |
| LaMP-4 | Generation | 12527 | 1925 | 2376 | $30.53 \pm 12.67$ | $9.78 \pm 3.10$ | $287.16 \pm 360.62$ | - |
| LaMP-5 | Generation | 9682 | 2500 | 2500 | $152.81 \pm 86.60$ | $9.26 \pm 3.13$ | $89.61 \pm 53.87$ | - |

## E  Baseline Details

We compare our proposed HYDRA with both non-personalized and personalized baselines. The non-personalized baselines consist of zero-shot gpt-3.5-turbo [30] and **ICL-Random** (In-Context Learning with Random items from user behavior history). On the other hand, the personalized baselines include **RAG** (Retrieval-Augmented Personalization) and **PAG** (Profile-Augmented Personalization). $k$ denotes the number of retrieved items from the user behavior history (or profiles). For all baselines, we follow the same prompt template in the LaMP benchmark [41] and employ gpt-3.5-turbo(1106) as the backbone black-box LLM. We utilize BM25 [39] as the default retriever for all experiments.

- **gpt-3.5-turbo** [30] processes only the user's query (*i.e.*, the input question) without leveraging the user's profile data.

- **ICL-Random** augments the user's query with random $k$ items from behavior history, enabling in-context learning to improve the capability of the original backbone LLM.

- **RAG** [41] augments the user's query with top $k$ retrieved items from the corresponding user's behavior history, following the retrieval-augmented personalization method in LaMP.

- **PAG** [38] augments the user's query with the user's profile summary generated by LLMs (gpt-3.5-turbo) which concludes the user's behavior patterns. We further combine PAG with $k$ retrieved items from the corresponding user's behavior history.

# F  Implementation Details

**Hardware and Software.** We conduct all black-box LLM personalization experiments on CPU: AMD(R) EPYC(R) 7702 64-Core Processor @ 1.50GHz and GPU: NVIDIA A100-SXM4-80GB using Python 3.10.13.

**Hyperparameter Configurations.** We set the maximum sequence length for a generated solution as $512$ tokens and the temperature as $1.0$ for flexibility in the generations of black-box LLMs, which serve as potential solution candidates. For other baselines, we maintain the temperature to $0$ to avoid potential instability in performance. During the black-box LLM adaption stage, we set $b = 8$ for the generation of intermediate candidates using `HYDRA`-Adapter. During the training stage, we set the learning rate to $2e - 5$, the global batch size to $64$, and the number of training epochs to 2 as default hyperparameter settings for all experiments. In addition, we employ `AdamW` [19] as the optimizer with a weight decay of 0.01. We include prompt details in Appendix I.

# G  Additional Experimental Results and Analysis

## G.1  Additional Analysis of Baselines in the Main Experimental Results

We conduct further analysis of the main experimental results of dominant approaches for black-box LLMs, specifically prompt-based methods, including RAG and PAG. Compared to the zero-shot setting in `gpt-3.5-turbo` [30], even a random selection of historical records from user behavior or profiles enhances the model performance in most tasks, suggesting that personalization contributes to improved performance with black-box LLMs. However, we also observe several instances of less optimal performance, particularly in LaMP-3. We hypothesize that this inconsistency may arise from the introduction of noise and irrelevant behavior patterns by the retrieved items, potentially complicating the understanding of user behavior patterns. Additionally, both RAG [41] and PAG [38] demonstrate marginal improvements over random selection, which may benefit from augmenting relevant behavior or profile information. The experimental results also consistently demonstrate that an increase in the number of retrieved items is positively correlated with improved performance, indicating the effectiveness of the retrieval-augmented framework in black-box LLM personalization. However, in comparison to `HYDRA`, RAG- and PAG-based methods still yield suboptimal results. This is primarily attributed to the presence of potentially noisy and irrelevant retrieved behavior patterns, as well as the absence of shared general knowledge across the entire user group.

## G.2  Additional Experimental Details for Sacle-up Analysis

Figure 7 reports additional experimental results of `HYDRA` in accuracy and F-1 with different numbers of selected history per user, indicating the robustness of `HYDRA` with additional user history.

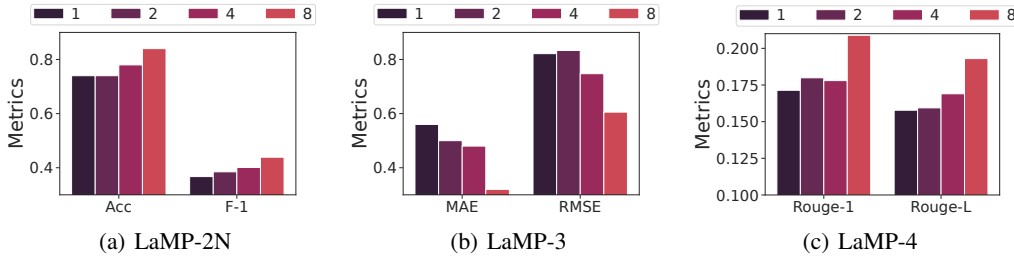

(a) LaMP-2N          (b) LaMP-3          (c) LaMP-4

Figure 7: Effect of users with different numbers of behavior history in `HYDRA`.

## G.3  Additional Experimental Details for User Behavior Shift

Following the setup of existing work [50], we utilize an encoder-only language model called DeBERTa-v3 [11] to encode the user's historical behaviors and the query into high-dimensional representations. We then compute the cosine similarity between the query and all historical items to evaluate their relevance. We select the top 20 items with the lowest relevance scores as irrelevant

user history. As depicted in Figure 5, our proposed HYDRA still outperforms the state-of-the-art baselines. This performance improvement may stem from the personalized reranker, which reranks the candidates based on their usefulness rather than just relevance. Additionally, the personalized adapter also adapts the model's generations to align with user preferences. In addition, prompt-based methods tend to underperform when there is a distribution shift between the few-shot demonstrations and user queries. This issue can become even more severe under personalized scenarios, where there might be a mismatch between the selected user history and the user query.

## G.4 Effect of Retrievers

We investigate the effect of different retrievers in Table 7. Under the black-box LLM personalization scenario, dense retrievers that capture semantic relevance (*e.g.*, Contriever [16]) achieve similar performances to sparse retrievers (*e.g.*, BM25 [39] in Table 1), which is aligned with observations in the existing works [40].

Table 7: Additional experimental results with Contriever [16] as the retriever ($k$=4).

| Dataset ($\rightarrow$) | LaMP-2N | | LaMP-3 | | LaMP-4 | | |
|---|---|---|---|---|---|---|---|
| Method ($\downarrow$) | Acc. $\uparrow$ | F-1 $\uparrow$ | MAE $\downarrow$ | RMSE $\downarrow$ | R-1 $\uparrow$ | R-L $\uparrow$ | BLEU $\uparrow$ |
| ICL-Random | 0.660 | 0.288 | 0.560 | 0.917 | 0.163 | 0.148 | 0.159 |
| RAG | 0.680 | 0.293 | 0.540 | 0.883 | 0.179 | 0.152 | 2.082 |
| HYDRA | **0.800** | **0.414** | **0.380** | **0.730** | **0.189** | **0.173** | **2.501** |

## G.5 Effect of Black-Box LLM Adapters

We evaluate the effectiveness of different black-box LLM adapters in Table 8. Due to the inaccessibility to model internal parameters, BBox-Adapter is the only black-box LLM adapter available, apart from our proposed HYDRA-Adapter. Based on Table 8, it can be observed that HYDRA-Adapter outperforms BBox-Adapter when applied under the same personalization settings (as discussed in Section 3.4). This result demonstrates the effectiveness of our adapter module design, which captures both global and user-specific knowledge to further adapt to personalized model outputs.

Table 8: Effect of different black-box LLM adapters ($k$=4).

| Dataset ($\rightarrow$) | LaMP-3 | | LaMP-4 | | | LaMP-5 | | |
|---|---|---|---|---|---|---|---|---|
| Method ($\downarrow$) | MAE $\downarrow$ | RMSE $\downarrow$ | R-1 $\uparrow$ | R-L $\uparrow$ | BLEU $\uparrow$ | R-1 $\uparrow$ | R-L $\uparrow$ | BLEU $\uparrow$ |
| Personalized BBox-Adapter [48] | 0.540 | 0.860 | **0.151** | **0.135** | **1.285** | 0.394 | 0.337 | 5.478 |
| HYDRA-Adapter | **0.420** | **0.762** | 0.145 | 0.118 | 1.137 | **0.409** | **0.355** | **5.816** |

## G.6 Additional Datasets

To further validate the effectiveness of HYDRA, we conduct extensive experiments on two additional widely used personalization datasets [28], MovieLens-1M and Recipe, focusing on predicting users' personal ratings for movies or recipes based on their historical rating patterns. The experimental results (Table 9) demonstrate that HYDRA outperforms the best-performing baselines by 8.0% on Movielens and 10.3% on Recipe, respectively.

## G.7 Efficiency Analysis

Assume that we have $N_{\text{train}}$ users in the training data and $N_{\text{test}}$ users in the test set. We adopt the transformer architecture, specifically the Longformer-base, as the reranker and the adapter base models. Consequently, the time complexity of all stages should be proportional to that of the attention mechanism in transformers, $O(dL^2)$, where $L$ indicates the sequence length and $d$ indicates the hidden dimension. Additionally, the training process will go through the transformer for $T$ epochs, while inference only requires one. For each user in the HYDRA-Reranker training data, we augment $M$ random historical records from the user's corresponding profile. The retriever then retrieves the

| Dataset ($\rightarrow$) | MovieLens-1M | | Recipe | |
|---|---|---|---|---|
| Method ($\downarrow$) | MAE $\downarrow$ | RMSE $\downarrow$ | MAE $\downarrow$ | RMSE $\downarrow$ |
| `gpt-3.5-turbo` | 0.780 | 1.208 | 0.880 | 1.149 |
| ICL-Random (k=1) | 0.880 | 1.2649 | 0.960 | 1.233 |
| ICL-Random (k=2) | 0.800 | 1.095 | 0.980 | 1.192 |
| ICL-Random (k=4) | 0.820 | 1.225 | 0.760 | 1.114 |
| RAG (k=1) | 0.800 | 1.166 | 0.980 | 1.192 |
| RAG (k=2) | 0.700 | 1.030 | 0.880 | 1.114 |
| RAG (k=4) | 0.640 | 1.000 | 0.820 | 1.049 |
| PAG (k=0) | 0.820 | 1.147 | 0.860 | 1.128 |
| PAG (k=1) | 0.760 | 1.010 | 0.800 | 1.063 |
| `HYDRA` | **0.600** | **0.908** | **0.720** | **0.938** |

Table 9: Experiments on two additional personalization datasets, MovieLens-1M and Recipe, focusing on predicting users' personal ratings for movies or recipes based on their historical rating patterns.

| Method | Mode | Time Complexity | LaMP-2N | LaMP-2M | LaMP-3 | LaMP-4 | LaMP-5 |
|---|---|---|---|---|---|---|---|
| HYDRA-Reranker | Training | $O(N_{\text{train}}(M^2+1)TL^2d)$ | 31m10s | 41m51s | 50m37s | 1h1m31s | 1h8m16s |
| HYDRA-Reranker | Fit New User | $O(N_{\text{test}}(M^2+1)TL^2d)$ | 18m8s | 21m17s | 25m36s | 33m52s | 31m25s |
| HYDRA-Reranker | Inference | $O(N_{\text{test}}L^2d)$ | 3m4s | 3m1s | 3m7s | 4m38s | 5m14s |
| HYDRA-Adapter | Training | $O(N_{\text{train}}k\bar{H}TL^2d)$ | 1h10m17s | 2h2m16s | 2h1m59s | 3h56m47s | 3h19m42s |
| HYDRA-Adapter | Fit New User | $O(N_{\text{test}}k\bar{H}TL^2d)$ | 28m15s | 1h7m27s | 1h19s | 2h23m10s | 1h59m2s |
| HYDRA-Adapter | Inference | $O(N_{\text{test}}kL^2d)$ | 4m16s | 4m17s | 4m17s | 5m53s | 5m59s |

Table 10: **Time complexity analysis with running time summary** on the LaMP benchmark.

top-$M$ ($M = 20$ by default) relevant historical records to form training samples. Thus, for each user, we collect $M^2 + 1$ training samples. For each user in the HYPER-Adapter training data, we consider all the user historical records. For each record, we leverage model randomness to generate $k$ ($k = 8$ by default) samples for the adapter to select. Consequently, we can have $\bar{H}k$ training samples per user, where $\bar{H}$ denotes the average number of histories per user.

# H  Case Studies

## H.1  Case Studies for Effectiveness of Reranker

Table 11 illustrates two observed cases during the reranking process. In Case 1, the retriever accurately retrieves the text in the optimal order, meaning that the top outputs are the most relevant to the target. In this scenario, the reranker successfully generates scores that align with the retriever, preserving the correct relevance. In Case 2, when the retriever fails to retrieve the text in the optimal order by relevance, the reranker assigns higher scores to the more relevant terms, thereby mitigating the incorrect retrieval ordering. This emphasizes the significance of HYDRA-reranker by offering an additional measurement dimension that is dedicated to assessing usefulness.

## H.2  Error Analysis

Table 12 presents three common error types observed in the reranker that account for most general errors: (1) **Saturated Scores**: These errors occur when the ranker outputs overconfident scores (near 1.0) regardless of different inputs. Consequently, it becomes challenging to distinguish the best output as the differences among the scores are minimal. (2) **Incorrect Scores**: These errors arise when the ranker assigns higher scores to less relevant terms. As a result, the likelihood of selecting the most suitable samples decreases, leading to suboptimal generation. (3) **Divergent Scores**: These errors occur when the ranker exaggerates the differences among the inputs. This may be attributed to the inputs being hard samples, where none of them are closer to the target compared to each other, making it difficult for the reranker to discern the most appropriate option.

Table 11: Case study for `HYDRA`-reranker. The "Target" column indicates the ground-truth categorization, while "Gen" represents the category of the retrieved text. "Score" is the score generated by the reranker, and "Order" denotes the retriever's ranking during the retrieval process.

| Case ID | Source | Retrieved text | Target | Gen | Order | Score |
|---------|--------|----------------|--------|-----|-------|-------|
| Case 1 | Eleven places will each get at least $1 million to reduce their jail population. | At least eight people have died at Hampton Roads Regional Jail in the past 17 months. | politics | politics | 1 | 0.3525 |
| | | But he still could be the Republican with the best shot at stopping her. | politics | crime | 2 | 0.3031 |
| | | "Some information that he might find embarrassing needs to get out. Just to be fair." | politics | crime | 3 | 0.2867 |
| Case 2 | The name speaks for itself. These are of the same high quality of all their products are and the sizing is true to the fit. | So far so good with this one. I would say this is definitely worth the extra expense compare to those oval shaped cream colored ones made out of aluminum that I am sure you have seen in numerous places ... | 5 | 4 | 1 | 0.4488 |
| | | These are okay, I have had better quality sheets in the past for less or the same price. These wrinkle easy but if that does not bother you then it is all cool. I do have problems keeping these on my bed, ... | 5 | 5 | 2 | 0.5200 |
| | | Sandstone coaster are the only way to go to absorb all the moisture from your glass. Good quality and worth your money to protect your tables. I have two sets of these and I recently started to use these, ... | 5 | 5 | 3 | 0.4384 |

# I  Prompts

Following the RAG-based framework [41] and the PAG-based framework [38], we have implemented the prompt design for the RAG and PAG-based baselines. The details of the prompt design can be found in Table 13 and Table 14, respectively. We follow the RAG framework in `HYDRA`. Additional examples of RAG framework for each task ($k = 2$) are available as follows.

```
RAG Prompt Demo for LaMP-2N (k=2)

the category for the article: "a vaccine but instead of fighting off disease it
    attracts dogs" is "entertainment", and
the category for the article: "The president, and many of his European counterparts,
    had condemned Trump as dangerous during his run." is "politics"
Which category does this article relate to among the following categories? Just
    answer with the category name without further explanation. categories: [women,
    religion, politics, style & beauty, entertainment, culture & arts, sports,
    science & technology, travel, business, crime, education, healthy living,
    parents, food & drink] article: The guitarist will not play with the band
    on its upcoming tour.
```

```
RAG Prompt Demo for LaMP-2M (k=2)

the tag for the movie: "In a dystopian future, a totalitarian regime maintains
    peace by subduing the populace with a drug, and displays of emotion are
    punishable by death. A man in charge of enforcing the law rises to overthrow
    the system." is "dystopia", and
the tag for the movie: "Cobb, a skilled thief who commits corporate espionage by
```

Table 12: Three common error types of the reranker. The "Target" column indicates the ground-truth categorization, while "Gen" represents the category of the retrieved text. "Score" is the score generated by the reranker.

| Error | Source | Retrieved text | Target | Gen | Score |
|-------|--------|----------------|--------|-----|-------|
| Saturated Scores | great quality and durability, also a great price, i would buy again, i really recommend to all my friends and to anyone, looking for this kind product, | Good price, durable, very good product, i bought this to make my son birthday cake, come out nice, and he love it it!! i would buy again, and recommend to any one. | 5 | 5 | 0.9657 |
| | | excellent shoes, great product, great price, fast delivery, unfortunately i had to return this product, because was ordered wrong size, but the return transaction with amazon was very prompt and smooth, i would buy again, over all i would recommend to all my friends and to anybody :) | 5 | 5 | 0.9427 |
| | | great quality and durability, also a great price, i would buy again, i really recommend to all my friends and to anyone, looking for this kind product, | 5 | 5 | 0.9692 |
| Incorrect Scores | Very nice, bristles are well seated. Love the wooden handle, feels nice in the hand, sturdy. | Love these looms, they are all the right sizes and I prefer them to the knifty knitter looms. | 5 | 4 | 0.7221 |
| | | This is a very nice eye mask. Works very well when kept in the freezer. Actually keeps out the light, too. | 5 | 5 | 0.7080 |
| | | It was a rough start. I almost put the book down. I had guessed the ending as soon as the 4th person showed up. Very fast read. | 5 | 5 | 0.7053 |
| Divergent Scores | "She said she missed when she would go out with her girlfriends and get dressed up!" | Anna Jones, 18, was sitting in a parked car with friends when she was shot. | women | style | 0.4016 |
| | | The reptile even received a kiss at the MTV Movie & TV Awards. | women | style | 0.7800 |
| | | "Wouldn't disrespect that queen like that," the comedian said of the backlash she received for an awkward bit she did at the ceremony. | women | style | 0.2905 |

Table 13: RAG prompt design for five LaMP tasks. Concat($\cdot$) concatenates the input strings in order, and PPEP($\cdot$) composes the prompt for each retrieved item from the profile. [INPUT] represents the task's input.

| Task | Per Profile Entry Prompt (PPEP) | Aggregated Input Prompt (AIP) |
|------|--------------------------------|-------------------------------|
| LaMP-2N | "the category for the article: "$P_i$[text]" is ""$P_i$[category]"" | concat([PPEP($P_1$), ..., PPEP($P_n$)], ", and "). [INPUT] |
| LaMP-2M | "the tag for the movie: "$P_i$[description]" is "$P_i$[tag]" | concat([PPEP($P_1$), ..., PPEP($P_n$)], ", and "). [INPUT] |
| LaMP-3 | $P_i$[score] is the score for "$P_i$[text]" | concat([PPEP($P_1$), ..., PPEP($P_n$)], ", and "). [INPUT] |
| LaMP-4 | "$P_i$[title]" is the title for "$P_i$[text]" | concat([PPEP($P_1$), ..., PPEP($P_n$)], ", and "). [INPUT] |
| LaMP-5 | "$P_i$[title]" is the title for "$P_i$[abstract]" | concat([PPEP($P_1$), ..., PPEP($P_n$)], ", and "). Following the given patterns [INPUT] |

```
    infiltrating the subconscious of his targets is offered a chance to regain
    his old life as payment for a task considered to be impossible: \"inception\",
    the implantation of another person's idea into a target's subconscious." is
    "sci-fi"
Which tag does this movie relate to among the following tags? Just answer with
    the tag name without further explanation. tags: [sci-fi, based on a book,
```

Table 14: Summarization prompt design for the five LaMP tasks. [INPUT] represents the task's input.

| Task | Prompt |
|------|--------|
| LaMP-2N | Look at the following past articles this journalist has written and determine the most popular category they write in. Answer in the following form: most popular category: <category> |
| LaMP-2M | Which tag does this movie relate to among the following tags? Just answer with the tag name without further explanation |
| LaMP-3 | Based on this user's past reviews, what are the most common scores they give for positive and negative reviews? Answer in the following form: most common positive score: <most common positive score>, most common negative score: <most common negative score> |
| LaMP-4 | Given this author's previous articles, try to describe a template for their headlines. I want to be able to accurately predict the headline given one of their articles. Be specific about their style and wording; don't tell me anything generic. |
| LaMP-5 | Given this author's previous publications, try to describe a template for their titles. I want to be able to accurately predict the title of one of the papers from the abstract. Only generate the template description, nothing else. |

```
comedy, action, twist ending, dystopia, dark comedy, classic, psychology,
fantasy, romance, thought-provoking, social commentary, violence, true story]
description: A ticking-time-bomb insomniac and a slippery soap salesman channel
primal male aggression into a shocking new form of therapy. Their concept
catches on, with underground \"fight clubs\" forming in every town, until an
eccentric gets in the way and ignites an out-of-control spiral toward oblivion.
```

__________ RAG Prompt Demo for LaMP-3 (k=2) __________

```
5 is the score for "Luckily, I am married. But I have already sent HIGH PRIORITY
    messages to all my single girlfriends that this book is MUST reading for
    understanding the male mind and knowing how to deal with it...all wrapped up
    in an energetic, yet sympathetic package (how nice that Matt seems to actually
    like women!). I especially enjoyed the tip boxes, and co-author Fadal's female
    wisdom. Fun!", and
5 is the score for "I received this book direct from the author. This has no effect
    on my review.\n\nThis is book 3. If you've read this far it means you love the
    series, and loved this book. You cannot read this one without reading book 1
    and 2. Otherwise you would be lost trying to follow what happens. I loved this
    book. I just wish there was an epilogue to show more of what happens after
    everything. I know there is a book just about Gabriel so maybe that's my follow
    up to the story. Happy reading!"
What is the score of the following review on a scale of 1 to 5? just answer with 1,
    2, 3, 4, or 5 without further explanation. review: Colin Dodds does it again
    with Another Broken Wizard. A genius at evoking a mood, scene, and character
    Dodd elegantly weaves all three together in this story with equal weight. I have
    long been a fan of Dodds' poetry, and this novel reaches many poetic points in
    both his use of language and the intoxicating (no pun intended) way he manages
    to paint scenes with words so that very specific moods and nuanced emotion come
    alive. Dodds is truly a wordsmith, and he has created a host of characters,
    especially the main protagonists, that will be very hard to forget - mostly
    because we know these people, either because they dwell within us or because
    they live among us. Besides, it a damn good noir story about one of those
    forgotten New England towns, and it's full of blood, guts, grit, love, and loss.
    Hard to put down - highly recommended.
```

__________ RAG Prompt Demo for LaMP-4 (k=2) __________

```
"Social Media Gone Awry: Tips for Teens to Stay Safe" is the title for "Here are a
    few tips to keep your teen safe when using the Internet and other web-based
    technologies. If you think it's an awkward conversation; you can hand them this
    blog to read.", and
"If You See Something, Please Do Something to Prevent Child Abuse" is the title for
    "Although the age of social media has dramatically lowered the threshold on
```

privacy standards, many adults are still reticent about reporting their
    suspicions about child abuse and neglect."
Generate a headline for the following article: That explains how the Confederate
    flag, contraception and clean water deregulation got linked to fighting
    mosquitoes, the senator said.

---
RAG Prompt Demo for LaMP-5 (k=2)
---

"Accurate Estimators for Improving Minwise Hashing and b-Bit Minwise Hashing" is
    the title for "Minwise hashing is the standard technique in the context of
    search and databases for efficiently estimating set (e.g., high-dimensional 0/1
    vector) similarities. Recently, b-bit minwise hashing was proposed which
    significantly improves upon the original minwise hashing in practice by storing
    only the lowest b bits of each hashed value, as opposed to using 64 bits. b-bit
    hashing is particularly effective in applications which mainly concern sets of
    high similarities (e.g., the resemblance >0.5). However, there are other
    important applications in which not just pairs of high similarities matter. For
    example, many learning algorithms require all pairwise similarities and it is
    expected that only a small fraction of the pairs are similar. Furthermore, many
    applications care more about containment (e.g., how much one object is contain
    by another object) than the resemblance. In this paper, we show that the
    estimators for minwise hashing and b-bit minwise hashing used in the current
    practice can be systematically improved and the improvements are most
    significant for set pairs of low resemblance and high containment.", and
"b-Bit Minwise Hashing for Large-Scale Linear SVM" is the title for "In this paper,
    we propose to (seamlessly) integrate b-bit minwise hashing with linear SVM to
    substantially improve the training (and testing) efficiency using much smaller
    memory, with essentially no loss of accuracy. Theoretically, we prove that the
    resemblance matrix, the minwise hashing matrix, and the b-bit minwise hashing
    matrix are all positive definite matrices (kernels). Interestingly, our proof
    for the positive definiteness of the b-bit minwise hashing kernel naturally
    suggests a simple strategy to integrate b-bit hashing with linear SVM. Our
    technique is particularly useful when the data can not fit in memory, which is
    an increasingly critical issue in large-scale machine learning. Our preliminary
    experimental results on a publicly available webspam dataset (350K samples and
    16 million dimensions) verified the effectiveness of our algorithm. For example,
    the training time was reduced to merely a few seconds. In addition, our tech
    can be easily extended to many other linear and nonlinear machine learning
    applications such as logistic regression. "
Generate a title for the following abstract of a paper: We propose skewed stable
    random projectionsfor approximating the \u03b1th frequency moments of dynamic
    data streams (0 < \u03b1 \ufffd 2). We show the sample complexity (number of
    projections) k = G 1 \u01eb2 log ' 2 \u03b4 \u00b4 , where G ! \u01eb 2
    log(1+\u01eb) = O (\u01eb) as \u03b1 ! 1, i.e., \u03b1 = 1 \u00b1 \ufffd with
    \ufffd ! 0. Previous results based on symmetric stable random projections(12,
    16) required G = non-zero constant + O(\u01eb), even when \ufffd = 0. The case
    \ufffd ! 0 is practically important. For example, \ufffd might be the \"decay
    rate\" or \"interest rate,\" which is usuall y small; and hence one might view
    skewed stable random projectionsas a \"generalized counter\" for estimating the
    total value in the future, taking in account of the effect of decaying or inter
    accruement. We consider the popular Turnstile data stream model. The input data
    stream at = (i, It) arriving sequentially describes the underlying signal A,
    meaning At(i) = At 1(i) + It, i 2 (1, D). We allow the increment It to be either
    positive (i.e., insertion) or negative (i.e., del etion). By definition, the
    \u03b1th frequency moment F(\u03b1) = PD i=1 |At(i)| \u03b1. Our method only
    requires that, at the time t for the evaluation, A

