# OpenReview forum: "HYDRA: Model Factorization Framework for Black-Box LLM Personalization"
_NeurIPS.cc/2024/Conference — NeurIPS 2024 poster_

### Official Review · Reviewer_AVA9 · 2024-07-11

**Soundness:** 3
**Presentation:** 3
**Contribution:** 3
**Rating:** 6
**Confidence:** 4

**Summary:**

The paper proposes HYDRA, a learning-based model factorization framework that captures both user-specific and shared behavior patterns to enable effective personalization within black-box LLMs. The framework involves training a reranker to prioritize the most useful information from top-retrieved relevant historical records, and training an adapter to align the output with individual user-specific preferences, thereby eliminating the reliance on access to inherent model parameters of black-box LLMs. The experimental results demonstrate that HYDRA outperforms existing state-of-the-art prompt-based methods by achieving an average relative improvement of 9.01% across five diverse personalization tasks in the LaMP benchmark.

**Strengths:**

1. The paper proposes a novel method for personalizing black-box LLMs by combining user-specific behavior with shared knowledge, which addresses limitations of previous methods.
2. The experimental results show that HYDRA achieves an average relative improvement of 9.01% over existing state-of-the-art prompt-based methods across five diverse personalization tasks in the LaMP benchmark.
3. The authors plan to release the code repository and model checkpoints, promoting transparency and reproducibility in future research.

**Weaknesses:**

1. The Algorithm Details in Appendix C indicate that HYDRA requires multiple stages of training. However, the paper does not provide an analysis of HYDRA's efficiency.
2. The paper only describes the user-specific head as a single layer of a feed-forward network, without providing details about the base model.
3. HYDRA-Adapter Inference utilizes an LLM to generate b candidates and selects the one with the highest score as the final answer. This approach substantially increases the inference time.
4. As the number of users increases, the number and parameters of user-specific heads also increase accordingly. This not only raises concerns about computational expenses but also introduces storage issues, potentially posing challenges for practical applications.
5. The effectiveness of HYDRA heavily depends on the quality and relevance of historical data, which may not always be available in the real world.

**Questions:**

1. Could you provide the time complexity analysis, training runtime, and inference time?
2. Could you provide a more detailed introduction of the base model?
3. HYDRA-Adapter Inference uses LLM to generate b candidates, then selects the one with the highest score as the final answer. Considering the inherent randomness in LLM generation, is this fair to baselines?
4. How does HYDRA perform in scenarios where there are significant disparities in user behavior history, such as some users having dense histories while others have sparse ones?

**Limitations:**

1. HYDRA-Adapter Inference utilizes an LLM to generate b candidates and selects the one with the highest score as the final answer. This approach substantially increases the inference time.
2. As the number of users increases, the number and parameters of user-specific heads also increase accordingly. This not only raises concerns about computational expenses but also introduces storage issues, potentially posing challenges for practical applications.
3. The effectiveness of HYDRA heavily depends on the quality and relevance of historical data, which may not always be available in the real world.

---

> ### Author Rebuttal · Authors · 2024-08-07
>
> Thank you for your detailed suggestions. Please find our responses below:
>
> > W1&Q1: Time complexity.
>
> **A:** We summarize (1) the time complexity for the different stages and (2) the time consumed for training and inference on 100 training users and 50 test users in **Table R3**. Please see **official comment** below for detailed definitions. It is important to note that although current calculations are based on centralized computing, the fitting process for new users and inference can be distributed to user-end machines. This allows for simultaneous updates of different head models. The potential of parallelism and distributed training can further enhance the efficiency of HYDRA.
>
> > W2&Q2: Base model.
>
> **A:** Both HYDRA-Reranker and HYDRA-Adapter utilize the lightweight LongFormer-Base (110M) as the backend language models, as illustrated in Section 4.1. We include more details of the base model in the updated manuscript.
>
> > W3: Inference time.
>
> **A:** We acknowledge that the generation and selection of candidate solutions may lead to an increase in inference time. However, to the best of our knowledge, an alternative solution for black-box LLM personalization may not exist. We will include a time complexity analysis (**Table R3**) in the updated manuscript and prioritize improving the inference efficiency as an important future undertaking. We will be more than happy to include if the reviewer could suggest any alternative to black-box LLM personalization for inference comparison.
>
> > W4: Computational expenses.
>
> **A:** We would like to clarify that our proposed HYDRA design is more cost-efficient compared to many state-of-the-art baselines, such as tuning the entire LLM for each user. The personalized header for each user only requires 0.6M parameters. The total number of parameters in 10000 user-specific head models is almost equivalent to recent popular LLMs, specifically LLaMA-3-8B. Additionally, the personalized heads for each user can be trained and deployed on devices, eliminating the need for additional storage in the data center. We acknowledge that as the number of users increases, storage and computational expenses may gradually become concerns. We can further reduce the number of personalized heads and then combine incoming user-personalized heads with them as bases.
>
> > W5: Historical data.
>
> **A:** We conducted experiments to study the impact of data quality and relevance on historical user data. Regarding the quality of historical data,
> - **When historical user data is sparse, in the case of a limited number of users**, we observe that HYDRA reaches over 90% of best performance with only 20% of training data (Figure 3).
> - **Similarly, in the case of limited user history length**, HYDRA demonstrates its robustness by consistently capturing user-specific preferences (Figure 3).
>
> Regarding the relevance of historical data, instead of retrieving the most relevant items in RAG/PAG, HYDRA employs a retrieve-then-rerank workflow to **rerank retrieved items based on their usefulness**, thereby enabling a more comprehensive capture of implicit and complex personalized information.
> - **When historical user data is noisy, in the case of user behavior shifts**, HYDRA continues to outperform state-of-the-art baselines, even when the query is not relevant to all historical records (Figure 5).
>
> HYDRA maintains its robustness and consistency, even when dealing with sparse or noisy historical user data, by leveraging shared knowledge acquired from the base model.
>
> > Q3: Randomness in LLM generation.
>
> **A:** We emphasize the proposed method exploits the inherent randomness of LLM generation in the HYDRA-adapter to achieve more personalized responses and improve model performance. It is important to note that **even equipped with the randomness in LLM generation, existing baselines cannot take advantage of the randomness due to their inherent design**. As indicated in **Table R4**, the inherent randomness in LLM generation leads to almost no change or a significant decline in performance in other baselines due to a lack of capability in effectively utilizing the diversity in generation, which guarantees a fair comparison with HYDRA.
>
> In addition, we conduct additional experiments to mitigate the potential influence of randomness by replacing the HYDRA-Adapter component with (1) HYDRA (random), which randomly selects the final answer from the generated candidates, and (2) HYDRA (SC), which utilizes self-consistency to select the most common answer from the generated candidates. Note that self-consistency cannot be leveraged in generation tasks as there are no definite answers in model generation (LaMP-4 and -5). We observe that HYDRA-Adapter still outperforms these two baselines that introduce randomness, thus demonstrating the validity of the HYDRA-Adapter design.
>
> > Q4: Significant disparities in user behavior history.
>
> **A:** We would like to clarify that the current random selection of training users includes significant disparities in the number of historical records, ranging from less than 20 to over 200. Thus, our experiments simulate real-world scenarios, where some users have dense histories while others have sparse ones. HYDRA has shown superior performance compared to all baselines.
>
> To consider more extreme cases, **we retrain HYDRA on a mixture of 50% users with the fewest interactions and another 50% users with the most interactions** (**Figure R1**). The experimental results demonstrate that HYDRA consistently outperforms existing baselines even under extreme cases. Compared to the previous random selection of training users, HYDRA achieves relatively lower performance due to the imbalance of training samples for dense users and sparse users. By leveraging the global information in shared parameters, knowledge can be effectively transferred from dense users to sparse users, thereby enabling further personalization through the utilization of sparse user-specific head models.

---

> ### Author Response · Authors · 2024-08-07
> **Additional Details of Time Complexity Analysis**
>
> We conduct time complexity analysis as follows:
>
> Assume that we have $N_{train}$ users in the training data and $N_{test}$ users in the test set. We adopt the transformer architecture, specifically the Longformer-base, as the reranker and the adapter base models. Consequently, the time complexity of all stages should be proportional to that of the attention mechanism in transformers, $O(dL^2)$, where $L$ indicates the sequence length and $d$ indicates the hidden dimension. Additionally, the training process will go through the transformer for $T$ epochs, while inference only requires one.
>
> - For each user in the HYDRA-Reranker training data, we augment $M$ random historical records from the user’s corresponding profile. The retriever then retrieves the top-$M$ ($M=20$ by default) relevant historical records to form training samples. Thus, for each user, we collect $M^2+1$ training samples.
>
> - For each user in the HYPER-Adapter training data, we consider all the user historical records. For each record, we leverage model randomness to generate $k$ ($k=8$ by default) samples for the adapter to select. Consequently, we can have $\bar{H}k$ training samples per user, where $\bar{H}$ denotes the average number of histories per user.
>
> Therefore, the time complexity for the different stages is as follows:
>
> | Method | Mode | Time Complexity |
> |:--------|:-------|:------------------|
> | HYDRA-Reranker | Training | $O(N_{train}(M^2+1)Td(L^2))$ |
> | HYDRA-Reranker | Fit New User | $O(N_{test}(M^2+1)Td(L^2))$ |
> | HYDRA-Reranker | Inference | $O(N_{test}d(L^2))$ |
> | HYDRA-Adapter | Training | $O(N_{train}\bar{H}kdT(L^2))$ |
> | HYDRA-Adapter | Fit New User | $O(N_{test}\bar{H}kTd(L^2))$ |
> | HYDRA-Adapter | Inference | $O(N_{test}kd(L^2))$ |
>
> We then proceed to empirically examine the efficiency with respect to training and inference time consumption on 100 training users and 50 test users:
>
> | Method | Mode | LaMP-2N | LaMP-2M | LaMP-3 | LaMP-4 | LaMP-5 |
> |:--------|:-------|:------------------:|:----------------------:|:----------:|:-----------:|:-----------:|
> | HYDRA-Reranker |  Training | 31m10s | 41m51s | 50m37s | 1h1m31s | 1h8m16s |
> | HYDRA-Reranker | Fit New User | 18m8s | 21m17s | 25m36s | 33m52s | 31m25s |
> | HYDRA-Reranker | Inference | 3m4s | 3m1s | 3m7s | 4m38s | 5m14s |
> | HYDRA-Adapter | Training | 1h10m17s | 2h2m16s | 2h1m59s | 3h56m47s | 3h19m42s |
> | HYDRA-Adapter | Fit New User | 28m15s | 1h7m27s | 1h19s | 2h23m10s | 1h59m2s |
> | HYDRA-Adapter | Inference | 4m16s | 4m17s | 4m17s | 5m53s | 5m59s |
>
> It is important to note that although current calculations are based on centralized computing, the fitting process for new users and inference can be distributed to user-end machines. This allows for simultaneous updates of different head models. The potential of parallelism and distributed training can further enhance the efficiency of HYDRA.

---

> > ### Comment · Reviewer_AVA9 · 2024-08-11
> > **Acknowledgement**
> >
> > I thank the authors for their further explanation, and I would like to keep my score.

---

> > > ### Author Response · Authors · 2024-08-12
> > > **Thank You**
> > >
> > > Dear Reviewer AVA9,
> > >
> > > Thank you very much for taking the time to review our rebuttal and offering insightful feedback. We will update our paper with the additional results and discussions.
> > >
> > > Best Regards,
> > > Authors

---

### Official Review · Reviewer_KhW9 · 2024-07-14

**Soundness:** 3
**Presentation:** 3
**Contribution:** 2
**Rating:** 6
**Confidence:** 4

**Summary:**

This paper provides a black-box LLM personalization framework that explores global and local knowledge from user’s historical behaviour through model factorization.

**Strengths:**

Strengths:

1.	The paper is straightforward. The method is reasonable
2.	This is good to see the authors provided many details including analyses and case studies in Appendix G and H
3.	Extensive experiments were conducted with implementation details and prompts were also provided.
4.	Model factorization is a good idea

**Weaknesses:**

Weaknesses:

1.	The performance is not convincing. For example, in Table 1, it’s interesting to see that ICL-Random seems to be perform on par with HYDRA (second-best baseline on LaMP-2M, LaMP-5), and even better than RAG / PAG baselines. It’s not clear to me why using random items from user behaviour history is doing much better; please give more details.
2.	From the design until experiments, the authors should provide more details (or I may missed it?) about the tasks that benefit by shared (global), the tasks that benefit individual (local) preference, and the tasks that benefit from both. I consider model factorization is a big factor, but not much explanation and analyses were given for that part. In that way, it will further strengthen the paper
3.	In addition, in Table 2, for example, HYDRA -P.-Adapter&Reranker has the same performance as HYDRA. Section 4.3 should provide more analyses and explanation about that.
4.	Even though the authors provided implementation details, I still believe the source code should be released or more stronger analyses should be given, given the performance results I mentioned above.

**Questions:**

Please refer to my concerns above.

**Limitations:**

The authors did provide details in Appendix A.

---

> ### Author Rebuttal · Authors · 2024-08-07
>
> Thank you for your detailed suggestions and comments. Please find the corresponding responses below:
>
> > W1: The performance is not convincing.
>
> **A:** We appreciate your thorough observations regarding the performance of ICL-Random in comparison to other baselines and HYDRA. We would like to explain the consistent observations, including (1) ICL-Random performs equally or better than the RAG and PAG baselines, and (2) the performance of RAG and PAG does not necessarily increase with the number of retrieved items (k).
>
> The worse performance of RAG/PAG than ICL-Random indicates **the complexity of the LaMP dataset and the personalization task**, which contains **implicit user preferences** that are NOT easily captured by straightforward relevance (e.g., RAG) or profile matching (e.g., PAG). In addition, RAG and PAG may **introduce noisy information through retrieval**, even retrieving a larger number of retrieved items (k) does NOT necessarily lead to better results.
>
> Instead of retrieving the most relevant items, HYDRA employs a retrieve-then-rerank workflow to **rerank retrieved items based on their usefulness**, thereby enabling a more comprehensive capture of implicit and complex personalized information. This highlights the significance of a personalization framework like HYDRA, which effectively identifies and utilizes complex patterns of user behavior. Despite the relatively strong performance of ICL-Random in LaMP-2M and -5, **HYDRA still consistently outperforms all baselines across all tasks, including ICL-Random**. We will provide a more comprehensive performance analysis in the revised manuscript.
>
> > W2: Additional details on the base model.
>
> **A:** Thank you for your valuable suggestions. In our initial submission, we included dataset and task details in Appendix D. In addition, we would like to emphasize the distinction between prediction and generation tasks (Table 5 in Appendix D) with regard to the advantage of global or local preference. **Prediction tasks, such as LaMP-2N, -2M, and -3, benefit more from global knowledge.** This is because these prediction tasks can utilize general patterns learned from the entire user base, which particularly benefits users with limited historical records (i.e., cold-start). On the other hand, **generation tasks, such as LaMP-4 and -5, benefit more from local preference**. This is because personalized heads focus on customizing outputs for individual users, which offers enhanced personalization for users with extensive historical records.
>
> The model factorization of HYDRA consists of two main components:
> - shared base model capturing global information across all users and
> - personalized heads capturing user-specific preferences.
>
> The integration of shared and personalized components provides
> - complementary information, with global knowledge serving as a solid knowledge foundation, while personalized heads refine outputs for individual users;
> - model flexibility, allowing the balance between global and personalized information based on the available user history; and
> - model robustness, mitigating overfitting to individual user patterns by grounding personalization in global knowledge.
>
> Ablation studies demonstrate the effectiveness of each component across both classification and generation tasks (Table 2). In the revised manuscript, we will ensure a clearer emphasis on the specific task details of global and local preference with model factorization.
>
> > W3: HYDRA -P.-Adapter&Reranker has the same performance as HYDRA.
>
> **A:** We here clarify that across all five tasks, the HYDRA-P.-Adapter&Reranker achieves similar performance compared to the full HYDRA model **only** in LaMP-2N. This is due to task specificity, as LaMP-2N is a prediction task that involves predicting news articles into one of 15 categories based on the journalist. Given an article written by a user, the model predicts its category using the user's history of articles and their corresponding categories. As categorized in our response to W2, **LaMP-2N is the easiest task that relies more on reasoning based on objective facts and requires less personalization**. It is also important to note that even for specific tasks like LaMP-2N, the integration of the adapter and reranker will not negatively impact the model's performance. The ablation study in Table 2 still demonstrates the effectiveness of each component. We will incorporate this expanded analysis into Section 4.3 in the updated manuscript.
>
> > W4: Code release.
>
> **A:** Thank you for your suggestion. We have **included the source code as a zip file in the supplementary material** in the initial submission and **will publicly release the code on a GitHub repository to ensure transparency and reproducibility**. As per your suggestion, we will enhance the updated manuscript by including more comprehensive experimental results and rigorous analyses, which will encompass:
> - In-depth analysis of model performance regarding baseline comparisons (in response to **W1** above), task details with model factorization (**W2**), and ablation studies (**W3**);
> - Extensive experimental results on two additional personalization datasets (**Table R1**);
> - Scale-up experiments with additional analysis on an increased number of users (ranging from 100 to 1000, **Table R2**);
> - Efficiency test with computational complexity analysis (**Table R3**);
> - Effect of inherent randomness in LLM generation with two additional randomness-injected baselines (**Table R4**);
> - Simulation experimental results with analysis of significant disparities in user behavior history, including 50% of the most active users and 50% of the most inactive users (**Figure R1**).
>
> Thank you again for your review! We hope our responses address your concerns. Please let us know if you have any further questions, and we will be happy to discuss them further.

---

> ### Author Response · Authors · 2024-08-13
> **A Gentle Reminder**
>
> Dear Reviewer KhW9，
>
> Thank you again for your valuable feedback. As we have mentioned in the general comments, we would like to kindly remind you that the author/reviewer discussion phase ends by Aug 13th. Aside from the responses we have offered previously, we have **also offered more comprehensive experimental results and rigorous analyses in the attached 1-page PDF file**. We sincerely hope that our responses have enhanced the paper's quality and addressed your concerns. If there are any additional suggestions or comments you would like to provide, please don't hesitate to share them. We look forward to engaging in a constructive discussion during the rebuttal phase. Thank you!
>
> Best Regards,
> Authors

---

> ### Comment · Reviewer_KhW9 · 2024-08-13
> **Increase the score**
>
> Thanks for your response in details. I read all the responses and revisit the paper. I'll increase the score to weak accept. On a side note, please add README file to the source code upon release for reproducibility. Thank you.

---

> > ### Author Response · Authors · 2024-08-13
> > **Thank You**
> >
> > Dear Reviewer KhW9,
> >
> > Thank you very much for taking the time to review our rebuttal and offering insightful feedback. We will update our paper with the additional results and discussions. In addition, we will provide a detailed README file to the source code upon release and a project page for detailed introduction.
> >
> > Many thanks again!
> >
> > Best Regards,
> > Authors

---

### Official Review · Reviewer_xKC3 · 2024-07-15

**Soundness:** 3
**Presentation:** 3
**Contribution:** 2
**Rating:** 6
**Confidence:** 4

**Summary:**

The paper introduces HYDRA, a model factorization framework designed to personalize large language models (LLMs) without modifying their internal parameters. HYDRA addresses the challenge of personalizing inherently opaque, black-box LLMs through a retrieval-augmented workflow. This method enhances personalization by using historical data to effectively capture user-specific preferences. HYDRA consists of two primary components: a personalized reranker that selects relevant information from the user's history, and a personalized adapter that tailors the LLM’s outputs to individual preferences. By integrating both global knowledge and local user behaviors, HYDRA improves personalization effectiveness. Extensive testing shows that HYDRA surpasses other state-of-the-art methods in personalization tasks, proving its ability to deliver tailored experiences without altering the fundamental model parameters.

**Strengths:**

Originality: HYDRA introduces a novel model factorization method to personalize black-box large language models (LLMs). This marks a significant advancement beyond traditional techniques that necessitate access to model parameters. Distinct from existing methods which depend on prompt design or fine-tuning, HYDRA employs a dual-component system comprising a reranker and an adapter. This system enhances personalization without the need to modify the LLM's internal parameters, offering effective personalization within the limitations of black-box models.

Clarity: The paper is well-organized and clearly describes the components and operation of HYDRA. The explanations are concise and straightforward. Additionally, the inclusion of figures and detailed descriptions of algorithmic steps offers a clear view of the model’s structure and function, which helps in understanding the model’s mechanics.

**Weaknesses:**

1) Limited Evaluation Metrics: The evaluation primarily emphasizes improvements in accuracy and benchmark performance. However, the use of the LAMP dataset to measure personalization is quite limited and fails to convincingly demonstrate effectiveness.
2) Dependency on High-Quality Data: HYDRA's performance is significantly dependent on the quality of historical user data. If the data is sparse or noisy, the functionality of both the reranker and adapter could be undermined. Exploring and integrating methods to effectively manage such data limitations is crucial.
3) Scalability Concerns: The discussion in the paper about the model's effectiveness covers only a limited number of users. Conducting scalability tests with much larger datasets and more diverse user bases could help in affirmatively validating the framework's efficiency on a larger scale.
4) User Privacy Concerns: HYDRA uses user-specific data, raising potential privacy issues.
5) Handling Dynamically Changing Preferences: How does HYDRA manage continuously evolving user preferences? Is there an efficient mechanism in place to update user models without extensive retraining?
6) Limitations of Reranker and Adapter: What are the constraints on the learning capacity and efficiency of the reranker and adapter modules? Are there situations where these components fail to achieve optimal results?
7) Integration Challenges: What are the main challenges in integrating HYDRA into existing large-scale systems, especially those that already have personalization frameworks in place?

**Questions:**

1) Limited Evaluation Metrics: The evaluation primarily emphasizes improvements in accuracy and benchmark performance. However, the use of the LAMP dataset to measure personalization is quite limited and fails to convincingly demonstrate effectiveness.
2) Dependency on High-Quality Data: HYDRA's performance is significantly dependent on the quality of historical user data. If the data is sparse or noisy, the functionality of both the reranker and adapter could be undermined. Exploring and integrating methods to effectively manage such data limitations is crucial.
3) Scalability Concerns: The discussion in the paper about the model's effectiveness covers only a limited number of users. Conducting scalability tests with much larger datasets and more diverse user bases could help in affirmatively validating the framework's efficiency on a larger scale.
4) User Privacy Concerns: HYDRA uses user-specific data, raising potential privacy issues.
5) Handling Dynamically Changing Preferences: How does HYDRA manage continuously evolving user preferences? Is there an efficient mechanism in place to update user models without extensive retraining?
6) Limitations of Reranker and Adapter: What are the constraints on the learning capacity and efficiency of the reranker and adapter modules? Are there situations where these components fail to achieve optimal results?
7) Integration Challenges: What are the main challenges in integrating HYDRA into existing large-scale systems, especially those that already have personalization frameworks in place?

---

> ### Author Rebuttal · Authors · 2024-08-07
>
> Thank you for your detailed suggestions and comments. Please find the corresponding responses as follows:
>
> > W1: Limited Evaluation Metrics.
>
> **A:** In line with previous personalization research [1]- [3], LaMP serves as **a standard personalization benchmark that has been widely used ** in evaluating personalization for LLMs. To further validate the effectiveness of HYDRA, we conduct extensive experiments on two additional widely used personalization datasets [4]**, focusing on predicting users' personal ratings for movies or recipes based on their historical rating patterns. The experimental results (**Table R1**) demonstrate that HYDRA outperforms the best-performing baselines by **8.0% on Movielens and 10.3% on Recipe**, respectively.
>
> > W2: Dependency on High-Quality Data.
>
> **A:** We conducted experiments to study the impact of data quality on historical user data, as illustrated in Figures 3 and 5.
> - **When historical user data is sparse, in the case of a limited number of users**, we observe that HYDRA reaches over 90% of best performance with only 20% of training data, as shown in **Figure 3 (a)-(c)**.
> - **Similarly, in the case of limited user history length**, HYDRA demonstrates its robustness by consistently capturing user-specific preferences, as shown in **Figure 3 (d)-(f)**.
> - **When historical user data is noisy, when user behavior shifts**, HYDRA continues to outperform state-of-the-art baselines, even when the query is not relevant to all historical records, as shown in **Figure 5**.
>
> HYDRA maintains its robustness and consistency, even when dealing with sparse or noisy historical user data, by leveraging shared knowledge acquired from the base model.
>
> > W3: Scalability Concerns.
>
> **A:** Our experimental setup, including the number of users, aligns with other recent studies in personalized language modeling (**e.g., 100 users in [1] and [2]**). To address your valid concern, we conduct additional scale-up experiments (**Table R2**) to evaluate HYDRA **with an increased number of users, increasing from 100 to 1000, across all five tasks**. Our findings from the scale-up experiments show that HYDRA maintains its performance advantages over baselines as the number of users increases.
>
> > W4: User Privacy Concerns.
>
> **A:** We acknowledge the potential privacy concerns associated with the personalization task, as it naturally entails using individual users' historical preferences to customize the generations of LLMs. We would like to highlight that **HYDRA does not introduce any additional privacy risks compared to existing baselines**. While we acknowledge the significance of data privacy, achieving zero data leakage may be beyond the scope of this current work.
>
> With the increasing demand for privacy preservation, HYDRA's modular design offers the flexibility to remove the RAG component, thus downgrading it to HYDRA-Adapter (see Table 2). HYDRA-Adapter is able to achieve a higher level of privacy preservation without transferring any user historical data compared to existing baselines. As a potential extension, we can enhance privacy by collecting data from users who have given their consent to share their information, considering them as "anchor points”, and then map them to the most similar anchor users (groups).
>
> > W5: Handling Dynamically Changing Preferences.
>
> **A:** While dealing with dynamic changes may be orthogonal to our current personalization task, HYDRA is intrinsically compatible with continual learning. It enables efficient updates to user models as new data becomes available and eliminates the need for extensive retraining, achieving the balance between performance and efficiency. Specifically, we can follow a periodic continual training pipeline to (1) initially train the base model and personalized heads using the available user data and (2) collect a sufficient amount of new user records to adapt to dynamically changing preferences. Moreover, given that the shared knowledge among the entire user group may not significantly change over a short period, it is feasible to update the base model at a relatively slower pace than the head model.
>
> > W6: Limitations of Reranker and Adapter.
>
> **A:** We acknowledge the importance of understanding these limitations on the learning capacity and efficiency. As elaborated in the Limitation Section, the process of creating labeled datasets for training, especially for the HYDRA-reranker, introduces additional computational costs. In addition, the performance of the HYDRA-adapter may degrade when faced with user preferences that significantly deviate from the patterns seen during training. We will incorporate (1) newly conducted scale-up experiments in response to W3, as well as (2) additional analysis on user behavior shift experiments in Figure 5, in order to provide a comprehensive discussion on the learning capacity and efficiency of the reranker and adapter modules in the updated manuscript.
>
> > W7: Integration Challenges.
>
> **A:** To the best of our knowledge, **there are currently no widely available large-scale personalization systems for black-box LLMs**. However, we believe that the integration of HYDRA would be relatively simple due to its modular design. We can achieve this by (1) Leveraging existing user data to train HYDRA's base model and a set of personal headers, and (2) Quickly adapting to new users by composing personalized headers using a combination of existing user headers while leveraging the knowledge gained from existing users.
>
> [1] Tan et al. "Democratizing large language models via personalized parameter-efficient fine-tuning." arXiv 2024.
> [2] Tan et al. "Personalized Pieces: Efficient Personalized Large Language Models through Collaborative Efforts." arXiv preprint arXiv 2024.
> [3] Tang et al. "Step-Back Profiling: Distilling User History for Personalized Scientific Writing." arXiv 2024.
> [4] Lyu et al. "LLM-Rec: Personalized Recommendation via Prompting Large Language Models." NAACL 2024.

---

> ### Author Response · Authors · 2024-08-12
> **A Gentle Reminder**
>
> Dear Reviewer xKC3，
>
> Thank you again for your valuable feedback. As we have mentioned in the general comments, we would like to kindly remind you that the author/reviewer discussion phase ends by Aug 13th. Aside from the responses we have offered previously, we have **also offered more comprehensive experimental results and rigorous analyses in the attached 1-page PDF file**. We sincerely hope that our responses have enhanced the paper's quality and addressed your concerns. If there are any additional suggestions or comments you would like to provide, please don't hesitate to share them. We look forward to engaging in a constructive discussion during the rebuttal phase. Thank you!
>
> Best Regards,
> Authors

---

### Author Rebuttal · Authors · 2024-08-07

Dear reviewers,

We sincerely appreciate the time and effort dedicated to evaluating our work. We have summarized the additional experiments and analyses conducted during the rebuttal phase, and we are committed to incorporating them in the revised manuscript.

Our newly added main experiments and analysis (attached PDF file) include:
- **Experiments on two additional personalization datasets** (**Table R1**). We demonstrate that HYDRA outperforms the best-performing baselines by 8.0% on MovieLens-1M and 10.3% on Recipe, respectively;
- **Scale-up experiments with additional analysis on an increased number of users** (ranging from 100 to 1000, **Table R2**). We evaluate HYDRA across all five tasks and show that HYDRA maintains its performance advantages over baselines as the number of users increases;
- **Time efficiency examination with computational complexity analysis** (**Table R3**). We summarize the time complexity for the different stages and the time consumed for training and inference on 100 training users and 50 test users ;
- **Effect of inherent randomness in LLM generation with two additional randomness-injected baselines** (**Table R4**). We show that even equipped with the randomness in LLM generation, existing baselines cannot take advantage of the randomness due to their inherent design;
- **Experiments with analysis of significant disparities in user behavior history**, including 50% of the most active users and 50% of the most inactive users (**Figure R1**). The experimental results demonstrate that HYDRA consistently outperforms existing baselines even under extreme cases.

We would like to further emphasize our main contributions as follows:
- We propose HYDRA, the ***first black-box*** LLM personalization framework that effectively mines user behavior history and adapts to user preferences for enhanced user experience;
- HYDRA integrates ***shared (global) knowledge*** from the base model and ***individual (local) preference*** from multiple user-specific heads through model factorization to deliver generalizable personalization; and
- HYDRA significantly ***outperforms existing personalization baselines*** across five diverse tasks in the LaMP benchmark, introducing one of the first learning-based solutions that achieves more effective adaptation to individual users in black-box LLMs.

Please find the point-to-point response with additional details in the following rebuttal section. We sincerely hope that our responses have enhanced the paper's quality and addressed your concerns. If you have any additional suggestions or comments, please don't hesitate to share them. We look forward to engaging in a constructive discussion during the rebuttal phase. Thank you again for your understanding and consideration.

Best Regards,
Authors

---

### Author Response · Authors · 2024-08-11
**A gentle reminder**

Dear Reviewers,

Thank you again for your valuable feedback. We would like to kindly remind you that the author/reviewer discussion phase ends soon. We sincerely hope that our responses have enhanced the paper's quality and addressed your concerns. If there are any additional suggestions or comments you would like to provide, please don't hesitate to share them. We look forward to engaging in a constructive discussion during the rebuttal phase. Thank you!

Best Regards,

Authors

---

### Decision · Program_Chairs · 2024-09-25

**Decision:**

Accept (poster)

**Comment:**

The paper proposes a model factorization framework for black-box LLM personalization. The proposed framework consists of a reranker to retrieve relevant user history records, and an adapter to align the output with user preferences. Both reranker and adapter are decomposed into a base model for capturing global knowledge and multiple user-specific heads for personalization.

The reviewers noted and mostly agreed on the novelty of the approach, clarify of the presentation and extensive experiments.

The reviewers raised concerns on dependency on high-quality data, handling dynamically changing user preferences, training and inference efficiency, clear description of the base model etc.. The authors answered these questions during rebuttal. The authors are strongly encouraged to incorporate the analysis and results provided during the rebuttal and any other changes/updates needed into the final version to address the raised issues and further improve the paper.

From the additional results, it seems the training and inference cost of the proposed model is very high, e.g., training on 100 users can take a few hours. The size of the proposed model is also very large and not scalable with number of users, e.g., the personalized header for one single user requires 0.6M parameters. Both efficiency and model size can be big concerns for practical use of the proposed model. These issues need to be addressed in either the revised version or in an extended version of the paper.